# ROYAL SOCIETY
# OPEN SCIENCE

nuclear chemistry/materials science/
environmental chemistry

radiocatalysis, composite catalyst,
decolorization rate, methyl orange

**Author for correspondence:**
Wenbao Jia
e-mail: jiawenbao@163.com

This article has been edited by the Royal Society of Chemistry, including the commissioning, peer review process and editorial aspects up to the point of acceptance.

# Irradiation-catalysed degradation of methyl orange using BaF$_2$–TiO$_2$ nanocomposite catalysts prepared by a sol–gel method

Yongsheng Ling[1,2], Guang Wang[1], Ting Chen[3,4],
Xionghui Fei[1], Song Hu[1], Qing Shan[1], Daqian Hei[1],
Huajun Feng[3,4] and Wenbao Jia[1,2]

[1]Department of Nuclear Science and Engineering, Nanjing University of Aeronautics and Astronautics, 211106 Nanjing, People's Republic of China
[2]Collaborative Innovation Center of Radiation Medicine of Jiangsu Higher Education Institutions, 215021 Suzhou, People's Republic of China
[3]School of Environment Science and Engineering, Zhejiang Gongshang University, Hangzhou 310012, People's Republic of China
[4]Zhejiang Provincial Key Laboratory of Solid Waste Treatment and Recycling, Hangzhou 310012, People's Republic of China

GW, 0000-0002-7497-0910

BaF$_2$–TiO$_2$ nanocomposite material (hereinafter called the composite) was prepared by a sol–gel method. The composite surface area, morphology and structure were characterized by Brunauer–Emmett–Teller method, X-ray diffraction analysis and a scanning electron microscopy. The results showed that BaF$_2$ and TiO$_2$ form a PN-like structure on the surface of the composite. Composites were used to catalyse the degradation of methyl orange by irradiation with ultraviolet light, γ-rays and an electron beam (EB). It was demonstrated that the composite is found to be more efficient than the prepared TiO$_2$ and commercial P25 in the degradation of methyl orange under γ-irradiation. Increasing the composite catalyst concentration within a certain range can effectively improve the decolorization rate of the methyl orange solution. However, when the composite material is used to catalyse the degradation of organic matter in the presence of ultraviolet light or 10 MeV EB irradiation, the catalytic effect is poor or substantially ineffective. In addition, a hybrid mechanism is proposed; BaF$_2$ absorbs γ-rays to generate radioluminescence

and further excites $TiO_2$ to generate photo-charges. Due to the heterojunction effect, the resulting photo-charge will produce more active particles. This seems to be a possible mechanism to explain γ-irradiation's catalytic behaviour.

# 1. Introduction

With the ongoing development in the textile printing and dyeing industry, a large amount of toxic and not easily degradable wastewater is continually being discharged into the environment. The textile printing and dyeing industry mainly produce organic wastewater, with azo dyes as a typical pollutant [1,2]. Therefore, the proper treatment of wastewater, containing azo dyes (e.g. methyl orange), is currently a central area of research in water treatment [3].

In wastewater treatment, UV degradation and ionizing radiation degradation are the promising methods, which have great potential for the conversion of photon energy into chemical energy to degrade the pollutants in water [4]. Studies have shown that a series of highly reactive particles (e.g. •OH, •H and $e_{aq}^-$) are produced after exposing the water to high-energy radiation (as shown in equation (1.1)) [5,6]. These particles can react with aqueous pollutants by means of several reactions (i.e. addition, substitution, electron transfer and bond cleavage) for pollutant removal and water purification (as shown in equations (1.2) and (1.3)) [7,8].

$$H_2O \rightarrow (2.7)e_{aq}^- + (2.8)\bullet OH + (0.55)\bullet H + (0.45)H_2 + (0.7)H_2O_2 + (2.7)H^+, \tag{1.1}$$

$$R + \bullet OH \rightarrow R\bullet \tag{1.2}$$

and
$$R\bullet + \bullet OH \rightarrow CO_2 + H_2O. \tag{1.3}$$

The values in parentheses in equation (1.1) are the radiochemical yield $G$ values (µmol $J^{-1}$) of each of the active particle generated at a pH of 6–8. In equations (1.2) and (1.3), R represents organic pollutants and R• represents intermediate products.

In the series of processes, the yield of active particles plays an important role. However, like conventional ultraviolet radiation and ionizing radiation, there are problems requiring large doses and long reaction times before producing sufficient active particles [9–11]. To this end, a catalyst is added to increase the yield of the active particles during the irradiation, thereby reducing the irradiation time and doses.

When $TiO_2$ (as a semiconductor material) is irradiated with UV, the active centre of $TiO_2$ is photo-activated and an electron/hole ($e/h^+$) couple is obtained [12–16]. The electron/hole pair further reacts with oxygen and water to produce superoxide radical ion ($O_2^{\bullet -}$) and hydroxyl radical (HO•), both of which are very reactive and strongly oxidizing to be capable of effectively catalysing the degradation of organic pollutants and saving reaction time. Although $TiO_2$ has been demonstrated to be an effective catalyst in the presence of UV radiation, it is unsuitable for use as a catalyst in the presence of high-energy and high-permeability ionizing radiation because its band gap is only 3.2 eV. To compensate for the deficiency of traditional $TiO_2$ in high-energy ionizing radiation catalytic oxidation, modifying $TiO_2$ through doping with certain materials that can use high-energy radiation has been considered. As a scintillator material, $BaF_2$ is one of the activated materials that are used as radioluminescent (RL) agents. RL is the phenomenon to produce luminescence in a material by the bombardment of γ-radiation or an electron beam (EB). The literature reveals that when $BaF_2$ is bombarded with high-energy ions, the electrons on the $Ba^{2+}$ (5p) band are excited to the conduction band to leave the holes, and the electrons on the $F^-$ (2p) valence band are transitioned to Ba (5p), which produces RL [17]. Therefore, $BaF_2$ can effectively absorb high-energy radiation and emit ultraviolet light of 220 and 315 nm, which is then used to excite $TiO_2$ for photocatalysis to produce more active particles [18–20].

In this study, $BaF_2$–$TiO_2$ composites were prepared by a sol–gel method and characterized by X-ray diffraction (XRD), scanning electron microscopy (SEM) and Brunauer–Emmett–Teller (BET) method. The results show that the composite material forms an interesting Ti–F–Ba boundary microstructure, which established similar p–n junction potential between $BaF_2$ and $TiO_2$ [21–23]. The SEM mapping of the surface of the material reveals the elemental distribution of Ba, Ti, O and F on the surface of the composite prepared by different $TiO_2$–$BaF_2$ doping ratios. In addition, methyl orange solution was irradiated by UV radiation, γ-ray radiation and EBs, and the catalytic activity of the composite for the degradation of methyl orange by these three types of radiation was studied. A possible mechanism of hybrid of RL and heterojunction (HJ) is proposed to illustrate this radiation catalytic behaviour.

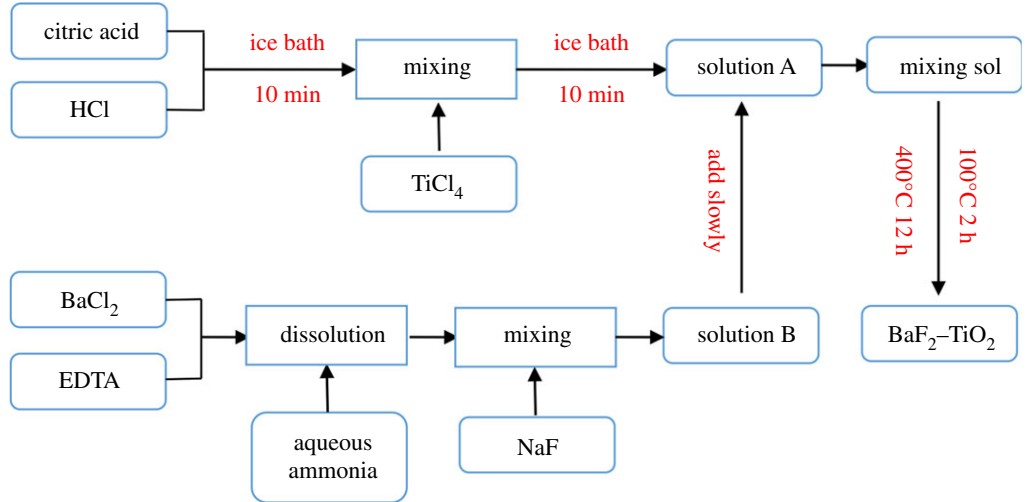

**Figure 1.** Schematic of the preparation the 'BaF$_2$–TiO$_2$' composite.

# 2. Material and methods

## 2.1. Chemicals and instruments

Reagent raw materials were TiCl$_4$ (greater than or equal to 99.5%, Sinopharm Chemical Reagent), HCl (36–38%, Sinopharm Chemical Reagent), citric acid (greater than or equal to 99.5%, Nanjing Chemical Reagent), ammonia (25–28%, Nanjing Chemical Reagent), BaCl$_2$ (greater than or equal to 99.5%, Nanjing Chemical Reagent), NaF (greater than or equal to 98%, Nanjing Chemical Reagent), ethylenediaminetetraacetic acid (EDTA, greater than or equal to 99.5%, Nanjing Chemical Reagent), quartz wool (1–3 μm, Nanjing Chemical Reagent), TiO$_2$ (P25, Aladdin Chemical Reagent) and methyl orange (AR, Nanjing Chemical Reagent); analytical balances (A1004B, Yoke Instrument), muffle furnace (SLR-1200, Shanghai Daheng Optics), high-temperature blast drying oven (XCT-0AS, Guangzhou Kenton), magnetic stirrer (H01-1G, Shanghai Mei Yingpu).

## 2.2. Catalyst preparation

The BaF$_2$–TiO$_2$ composite was prepared by a sol–gel method (figure 1) [24–33]. Briefly, 60 ml of 1 mol l$^{-1}$ HCl was mixed with 7.5 g of citric acid in a crucible and stirred until the solution became clear. In an ice bath, 10 ml of TiCl$_4$ was then added dropwise over the course of 10 min, and the solution (hereafter referred to as solution A) was stirred for an additional 10 min. The reaction process of the solution A is shown in equations (2.1)–(2.3). To prepare the solution B, 18 g of BaCl$_2$ and 27.5 g of EDTA were simultaneously dissolved by stirring in 245 ml of 2 mol l$^{-1}$ ammonia water (35 ml of concentrated ammonia) until the solution was clear, and 7.5 g of NaF was added. The reaction process of the solution B is shown in equations (2.4) and (2.5). The solution B was then slowly mixed with the solution A, and the mixture was magnetically stirred for 1 h to obtain a mixed sol. The product was then dried at 100°C for 2 h and calcined at 400°C for 12 h. The obtained powder was washed with distilled water until no Cl$^-$ was detected. Finally, the powder was dried in an oven at 105°C to obtain the BaF$_2$–TiO$_2$ material.

$$TiCl_4 + H_2O \rightarrow TiOH^{3+} + H^+ + 4Cl^-, \tag{2.1}$$
$$TiOH^{3+} \rightarrow TiO^{2+} + H^+, \tag{2.2}$$
$$TiO^{2+} + H_2O \rightarrow TiO_2 + 2H^+, \tag{2.3}$$
$$NH_3 \cdot H_2O + NaF \rightarrow NH_4F \tag{2.4}$$
and $$NH_4F + 2BaCl_2 \rightarrow BaF_2 + 2NH_4Cl. \tag{2.5}$$

Catalysts with different BaF$_2$ and TiO$_2$ contents (i.e. $C_{BaF_2} : C_{TiO_2} = 0.35, 0.75, 1.5$) were synthesized by changing the amount of the solution B added to the solution A. Composite catalysts are named $X$-BaF$_2$–TiO$_2$, where $X$ (0.35, 0.75 and 1.5) corresponds to the weight ratio of BaF$_2$ to TiO$_2$ in the material.

Additionally, neat TiO$_2$ and BaF$_2$ were obtained through a similar preparation method. In the following description, synthesized TiO$_2$ will be called 'TiO$_2$', while 'P25' will be used to designate the commercial TiO$_2$ obtained from Aladdin Chemical Reagent.

## 2.3. Characterization and analysis methods

The specific surface area and pore size distribution of the catalyst were determined by N$_2$ adsorption isotherm at 77 K, using a Micromeritic ASAP2460 instrument with 6 h degassing time at 200°C.

The crystal form of the catalyst was determined by XRD analysis performed on a Rigaku UItima IV diffractometer. With CuK$\alpha$ illumination, the scanning angle range was 10–80° (2$\theta$), the scanning step was 0.02° and the operating voltage and current were 40 kV and 40 mA, respectively. The crystallite size of the TiO$_2$ and BaF$_2$ was calculated using Scherrer's equation as follows [34,35]:

$$D = \frac{K\lambda}{\beta\cos\theta},$$  (2.6)

where $D$ is the crystal size of the catalyst, $K$ is a dimensionless constant, $\lambda$ is the wavelength of the X-ray, $\beta$ is the full width at half maximum (FWHM) of the diffraction peak and $\theta$ is the diffraction angle.

The morphology and elemental distribution of the catalyst were characterized by SEM using a SU8010 high-resolution field emission scanning electron microscope. The voltage was 10 keV, and the catalyst was observed at 5000 magnification. The obtained SEM image was analysed with the ImagePro Plus 6.0 software (Media Cybernetics, Inc., The Netherlands) program to determine the catalyst particle size [36]. Additionally, the two-phase interfaces of the composite catalyst were scanned to obtain the two-phase composition on the catalyst.

The absorbance of the methyl orange solution was scanned by ultraviolet–visible spectroscopy using an L5-type UV–visible spectrophotometer (Prisma, Shanghai). The scanning wavelength range was 350–600 nm, the scanning speed was set to medium and the scanning wavelength interval was set to 0.5 nm. The decolorization rate ($\eta$) (i.e. percentage reduction in colour value before and after irradiation) of methyl orange was selected as the index to investigate the effect of the composite on the photocatalytic decolorization of methyl orange. The absorbances of the methyl orange solutions $A_0$ and $A_1$ before and after irradiation, respectively, were measured, and the decolorization rate of methyl orange was calculated according to the following equation:

$$\eta = \frac{A_0 - A_1}{A_0} \times 100\%.$$  (2.7)

## 2.4. Catalytic degradation of methyl orange

The photocatalytic degradation of methyl orange was carried out in a 100 ml double-layer reaction flask using a mercury lamp as the ultraviolet light source with the reaction bottle connected to cooling water. Beginning with a 20 mg l$^{-1}$ methyl orange solution, the catalyst (P25, BaF$_2$, TiO$_2$, or BaF$_2$–TiO$_2$) was added with magnetic stirring according to the TiO$_2$ concentration gradient (i.e. the same TiO$_2$ content was used for each catalyst, and the mass concentrations BaF$_2$ and TiO$_2$ were identical). By stirring the solution, the catalyst was distributed uniformly. Initially, the methyl orange solution was magnetically stirred in the dark for 30 min to obtain an adsorption–desorption equilibrium between the dye and the catalyst. The mercury lamp was then turned on and the solution was exposed to UV light for 60 min.

Two samples were acquired for each analysis: the first after adsorption in the dark, and the second after the UV irradiation was complete. The samples were filtered through a 2 µm organic phase filter and stored in the dark. Finally, the absorbances of the samples were recorded simultaneously.

The γ-ray irradiation catalytic degradation of methyl orange was carried out using a $^{60}$Co source with a dose rate of 0.69 kGy h$^{-1}$. The prepared 20 mg l$^{-1}$ methyl orange solution was placed in several 20 ml irradiation bottles, and an equal amount of quartz wool was weighed into an irradiation bottle for dispersing the catalyst. The catalyst (P25, BaF$_2$, TiO$_2$ or BaF$_2$–TiO$_2$) was then added to the irradiation bottle according to the TiO$_2$ mass concentration gradient (i.e. the same TiO$_2$ content was used for each catalyst, and the mass concentrations BaF$_2$ and TiO$_2$ were identical).

One set of samples contained three samples, two of which were placed in the source chamber for 60 min. Another sample set was kept in dark conditions. After irradiation, all samples were tested for absorbance. The absorbance of each sample was obtained by subtracting the absorbance of the irradiated sample from the absorbance of the sample stored in the dark. In this way, the decolorization rate of each sample set of methyl orange solutions was calculated.

In the experiment of EB irradiation degradation of methyl orange solution, since the EB energy emitted by the electron accelerator is 10 MeV, the lowest absorbed dose (the amount of absorbed sample in a circle) after irradiation of the sample is 2 kGy. However, the 20 mg l$^{-1}$ methyl orange solution will be completely degraded at the lowest dose (2 kGy), which does not reflect the ability of the catalyst to catalyse the degradation of methyl orange, so we increase the concentration of methyl orange solution to 50 mg l$^{-1}$. Although we increase the solubility of the solution, it still belongs to the dilute aqueous solution, and does not affect the yield of unit dose of active particles, the experimental result is still very reliable [37]. In the experiment, the sample treatment method was identical to that used for the γ-irradiation experiments, and the configured sample was placed on the accelerator rail for one rotation (2 kGy r$^{-1}$). The absorbance of the irradiated sample was subtracted from the absorbance of the sample without EB irradiation to give the final absorbance change of each sample, which was then used to calculate the decolorization rate of the methyl orange solution.

# 3. Results and discussion

## 3.1. Characterization of the catalyst

XRD patterns of the prepared BaF$_2$ TiO$_2$ and BaF$_2$–TiO$_2$ samples are shown in figure 2. BaF$_2$ is of Frankdicksonite-type composition; its crystal form is cubic, as indicated by consistency with the standard diffraction peaks in JCPDS card no. 01-001-0533. The (101), (112), (200), (105), (211), (204), (116), (220) and (215) crystal faces of TiO$_2$ are all anatase crystal forms, as indicated by consistency with the standard diffraction peaks in JCPDS card no. 01-071-1167. The BaF$_2$–TiO$_2$ composites have almost all 2θ values of BaF$_2$ and TiO$_2$, but the peak intensities are different. It is observed that the peaks belonging to BaF$_2$ gradually increased with the increase in BaF$_2$ content for the composite catalysts. In addition, according to equation (1.1), the crystallite sizes of BaF$_2$ and TiO$_2$ are 38 and 27 nm, respectively.

Figure 3 shows the adsorption–desorption isotherm of the composite catalyst. It can be seen from figure 3 that all the samples have a typical type IV isotherm, indicating that the composite catalyst forms a mesoporous structure. According to the results, the adsorption capacity of 0.75-BaF$_2$–TiO$_2$ and 0.35-BaF$_2$–TiO$_2$ is similar. In table 1, the results from BET surface area measurements for the composite catalysts are given. As shown in table 1, specific surface area and pore size of TiO$_2$ were 59.3 m$^2$ g$^{-1}$ and 19.3 nm, while these values were 46.04–13.83 m$^2$ g$^{-1}$ and 23.1–30.17 nm for the composite catalyst, changing with the initial BaF$_2$ to TiO$_2$ ratios from 0.35 to 1.5, respectively. Overall, there is a decrease in the surface area when compared with that of the neat TiO$_2$. This indicated that nanophase TiO$_2$ particles were only embedded onto the surface of the BaF$_2$ substrates and the introduction of TiO$_2$ onto the surface of BaF$_2$ results in certain reduction in the specific surface area. The interface between these TiO$_2$ and BaF$_2$ should be the major reaction site to the catalytic reaction.

The morphologies of BaF$_2$, TiO$_2$ and composite catalysts were revealed by SEM investigation, and the SEM images of BaF$_2$, TiO$_2$ and representative composite catalyst (0.75-BaF$_2$–TiO$_2$) are shown in figure 4a–d. As seen, the BaF$_2$–TiO$_2$ composite particles are more regular and have a more defined shape than the prepared BaF$_2$ and TiO$_2$; the TiO$_2$ and BaF$_2$ particles in the composite catalyst are layered. The SEM image was processed using ImagePro Plus software, which showed that the particle size of BaF$_2$–TiO$_2$ composite (175–200 nm) is smaller (256–312 nm size) than that of BaF$_2$. Therefore, the TiO$_2$ particles hinder the aggregation of the BaF$_2$ particles. In addition, due to the difference in the band gap between TiO$_2$ and BaF$_2$, the synthesized composite forms a PN-like structure (similar to a bridge) between BaF$_2$ and TiO$_2$ compared to pure TiO$_2$ crystal. The synthesized composite can induce specific electron transfer processes, improve charge separation efficiency, produce more active particles and achieve better catalytic effects.

SEM mapping analyses were carried out to confirm the presence of TiO$_2$ and BaF$_2$ on the surface of composite catalysts. Typical SEM spectral images of 0.75-BaF$_2$–TiO$_2$ are presented in figure 5a–d, which are corresponding to the distribution of elements in the area of figure 4d. Figure 5a–d expressly confirms the presence of Ti, O, Ba, F and the elemental distribution of Ti, O or Ba, F is essentially the same. Figure 5e shows the specific content of the surface of composite catalyst; the relative content of elemental Ti is 44.3% and that of Ba is 19.7%. In addition, the SEM image has indicated that the surface of the composite catalyst forms a composite structure consisting of two phases. In order to explore the composition of the two phases, two points (A and B) were selected in figure 4d for SEM

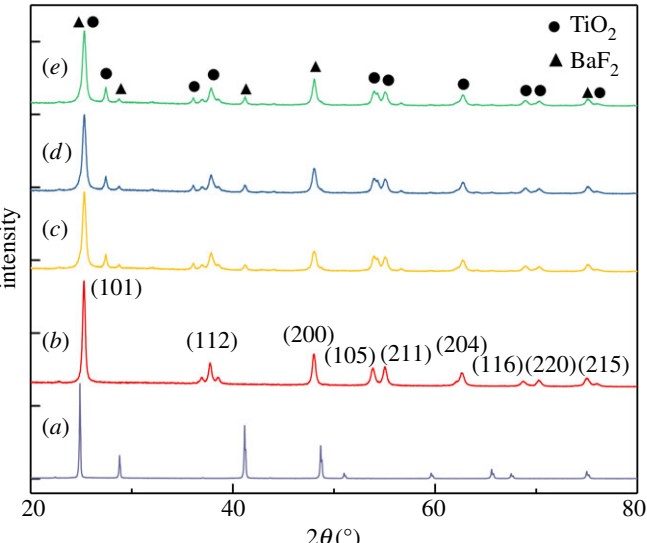

**Figure 2.** XRD patterns of (a) $BaF_2$, (b) $TiO_2$, (c) $BaF_2$–$TiO_2$ (0.35-$BaF_2$–$TiO_2$), (d) $BaF_2$–$TiO_2$ (0.75-$BaF_2$–$TiO_2$) and (e) $BaF_2$–$TiO_2$ (1.5-$BaF_2$–$TiO_2$).

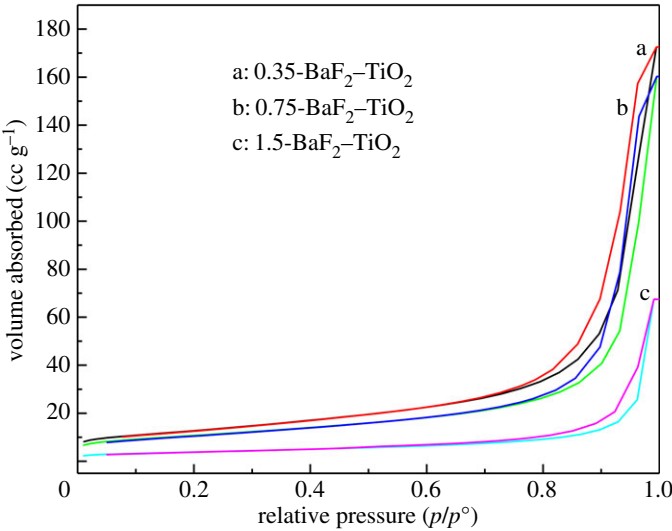

**Figure 3.** $N_2$ adsorption–desorption isotherm and pore size distribution of composite catalysts.

**Table 1.** Surface properties of catalysts.

| catalyst | BET surface area ($m^2\ g^{-1}$) | pore size (nm) |
|---|---|---|
| $TiO_2$ | 56 | 11.5 |
| 0.35-$BaF_2$–$TiO_2$ | 46.04 | 23.1 |
| 0.75-$BaF_2$–$TiO_2$ | 38.38 | 25.8 |
| 1.5-$BaF_2$–$TiO_2$ | 13.83 | 30.17 |

mapping analysis. The SEM mapping spectrum is presented in figure 5f,g, the point A consists mainly of Ba and F elements, and the point B consists mainly of Ti and O elements. The elemental composition of each point is almost the same as the elemental mass ratio of $TiO_2$ or $BaF_2$. Combined with the XRD patterns, the two phases that make up the composite catalyst are $TiO_2$ and $BaF_2$. As for other composite catalysts, their spectra are similar to those of 0.75-$BaF_2$–$TiO_2$, but the peak intensities are different due to different $BaF_2$ contents.

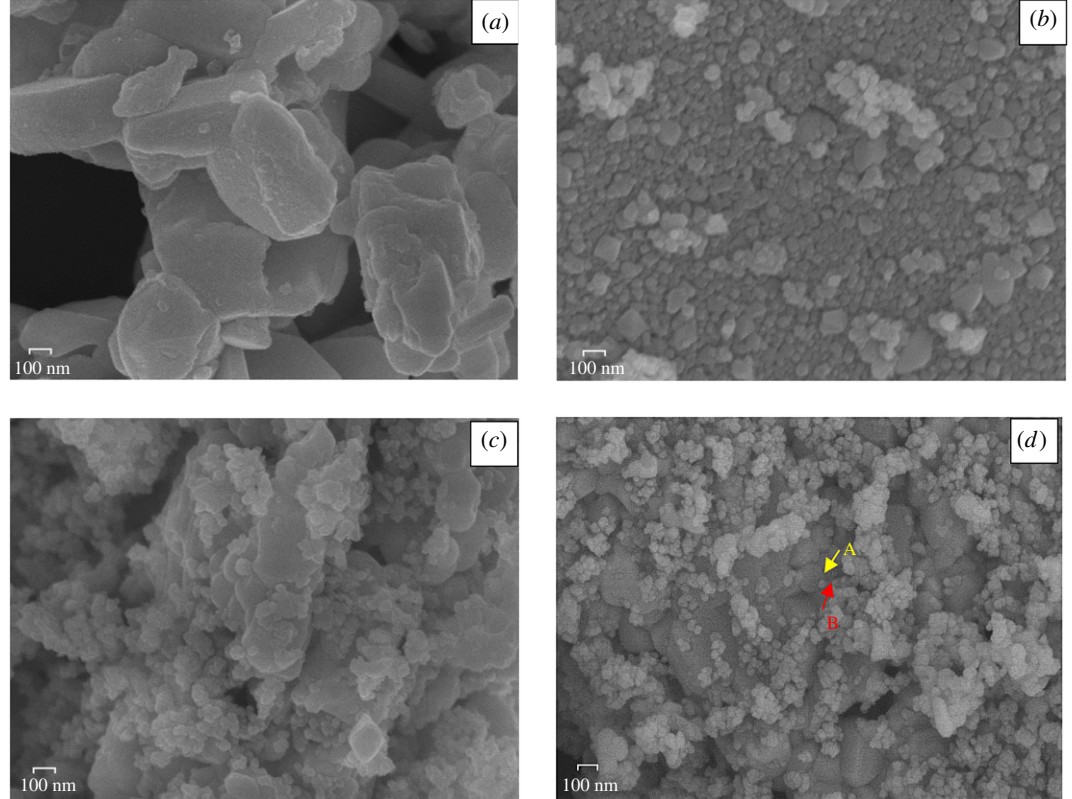

**Figure 4.** SEM micrographs of (a) $BaF_2$, (b) $TiO_2$ and (c,d) 0.75-$BaF_2$–$TiO_2$.

Since the concentration of the catalyst in the next catalytic experiment is based on the concentration of $TiO_2$ (i.e. the concentration of $TiO_2$ in the composite catalyst added the same as the concentration of pure $TiO_2$), it is necessary to know how much composite catalyst is required per gram of $TiO_2$. Table 2 lists the total mass required when different composite catalysts contain 1 g of $TiO_2$.

## 3.2. Catalytic degradation of methyl orange

### 3.2.1. Catalyst adsorption of methyl orange

The 1 h adsorption capacity of the different catalysts for methyl orange was tested in the dark using a $20~mg~l^{-1}$ methyl orange solution as a solvent. The experimental results are shown in figure 6 (for the sake of clearer images, figure 6 only shows the UV–visible spectrum of 0.75-$BaF_2$–$TiO_2$). P25 has the strongest adsorption capacity, and $BaF_2$ has almost no adsorption capacity as a catalyst. As for the composite catalyst, its adsorption capacity is between $BaF_2$ and $TiO_2$. 0.35-$BaF_2$–$TiO_2$ has the strongest adsorption capacity, and 1.5-$BaF_2$–$TiO_2$ has the lowest adsorption capacity. This is consistent with the $N_2$ adsorption results of figure 3. In general, the adsorption of methyl orange solution by the catalyst was small within 1 h. However, for the correctness of the data, the results of all experiments have deducted the adsorption of methyl orange by the catalyst.

### 3.2.2. Ultraviolet photocatalytic degradation of methyl orange

Using a mercury lamp as the ultraviolet light source, the catalytic effects of P25, $BaF_2$, $TiO_2$ and 0.35-$BaF_2$–$TiO_2$ on the degradation of methyl orange solutions were investigated under the same conditions; the catalyst concentration was $1~g~l^{-1}$, the 0.35-$BaF_2$–$TiO_2$ concentration was $1.35~g~l^{-1}$ and the $TiO_2$ concentration in the composite catalyst was $1~g~l^{-1}$. The UV–visible spectrum and decolorization rate of the methyl orange solution after UV irradiation for 1 h are shown in figures 7 and 8, respectively. As seen, the decolorization rate of the methyl orange solution irradiated with pure ultraviolet light was only 4.93%, indicating that pure ultraviolet light has little effect on the degradation of methyl orange. With the addition of $BaF_2$ as a catalyst, the decolorization rate of methyl orange decreased to 3.25%, demonstrating that $BaF_2$ is not suitable for use in UV-catalysed

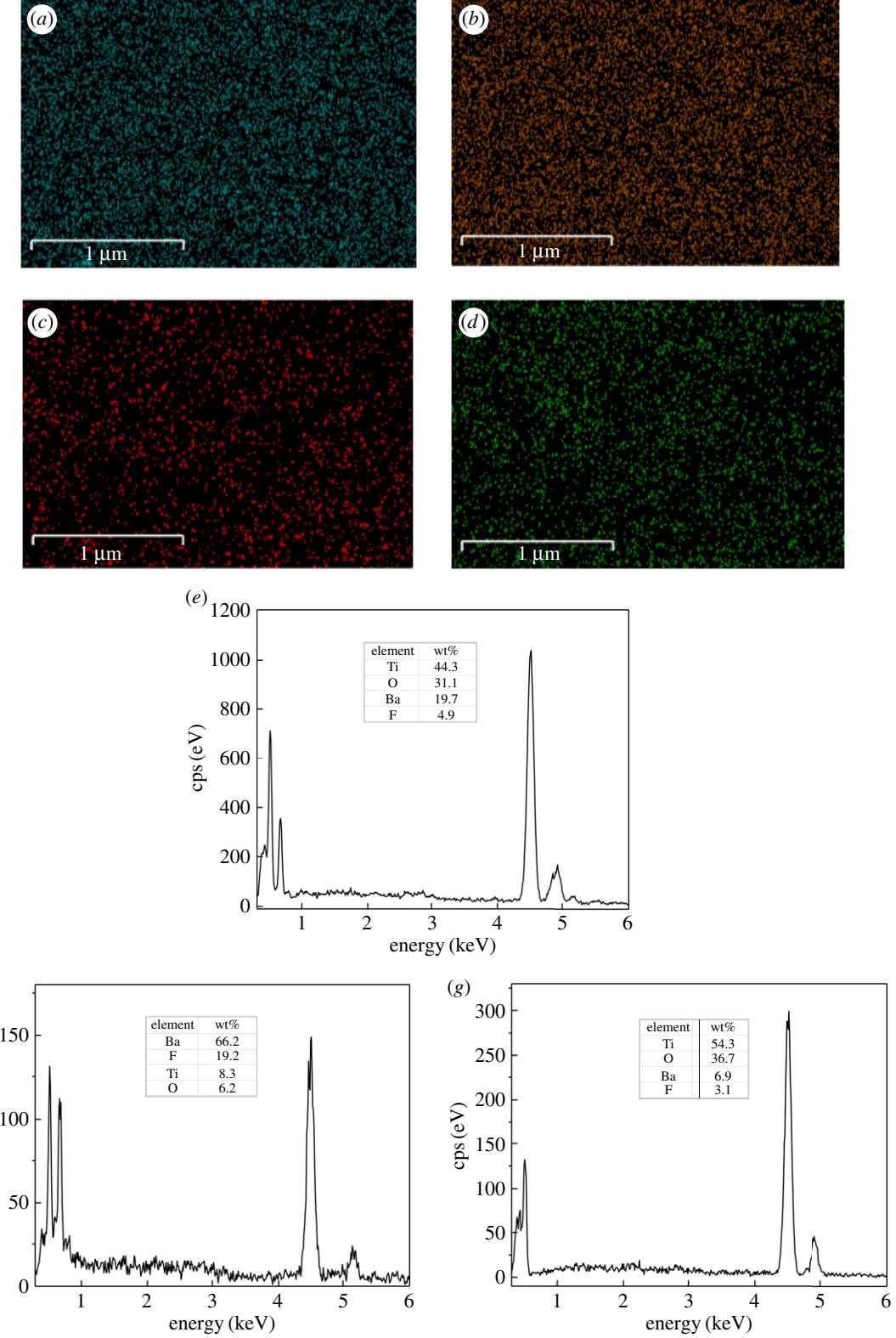

**Figure 5.** SEM mapping of 0.75-$BaF_2$–$TiO_2$. (*a*) Ti elemental distribution; (*b*) O elemental distribution; (*c*) Ba elemental distribution; (*d*) F elemental distribution; (*e*) elemental map of the catalyst surface; (*f*) elemental map of point A; (*g*) elemental map of point B.

degradation of methyl orange. This may result because the absorption of ultraviolet light by the methyl orange solution is hindered when $BaF_2$ particles are distributed in the solution, resulting in a decrease in the decolorization rate of the methyl orange solution. By comparison, the decolorization rate of the methyl orange solution reached 47.64% and 48.78%, respectively, when $TiO_2$ and P25 were added,

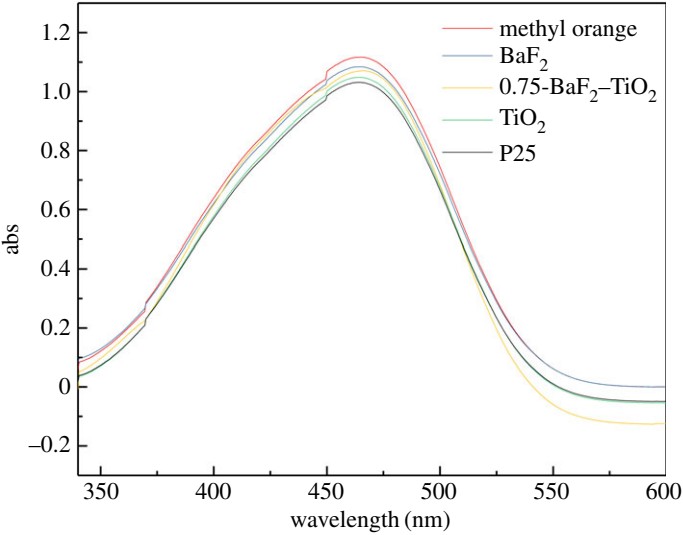

**Figure 6.** Ultraviolet–visible spectra of methyl orange solution adsorption by the different catalysts ($C_{BaF_2}$, $C_{TiO_2}$, $C_{P25}$: 1 g l$^{-1}$; $C_{0.75-BaF_2-TiO_2}$: 1.75 g l$^{-1}$).

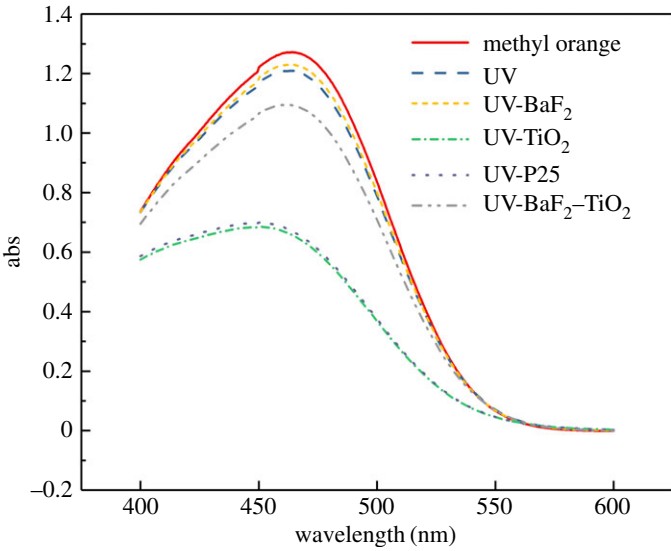

**Figure 7.** Ultraviolet–visible spectra before and after ultraviolet light irradiation of methyl orange solution with different catalysts ($C_{BaF_2}$, $C_{TiO_2}$, $C_{P25}$: 1 g l$^{-1}$; $C_{0.35-BaF_2-TiO_2}$: 1.35 g l$^{-1}$).

**Table 2.** Different composite catalysts Ti/Ba and mass of composite catalysts required for 1 g of TiO$_2$.

| catalyst | Ti/Ba | $m_{BaF_2-TiO_2}$ (g$^{-1}$) |
| --- | --- | --- |
| 0.35-BaF$_2$–TiO$_2$ | 6 | 1.35 |
| 0.75-BaF$_2$–TiO$_2$ | 3 | 1.75 |
| 1.5-BaF$_2$–TiO$_2$ | 1.5 | 2.5 |

indicating an almost identical photocatalytic ability of these catalysts. Combined with the above characterization results, these results further demonstrate that our synthesized TiO$_2$ is our desired crystal form. When the 0.35-BaF$_2$–TiO$_2$ composite was added as a catalyst, the decolorization rate of methyl orange solution was only 19.3%, which is lower than that of pure TiO$_2$. A possible reason for this is limited penetration of the ultraviolet light into the solution. The presence of BaF$_2$ hinders UV absorption by TiO$_2$, resulting in a rather weak photocatalytic ability of the prepared composite

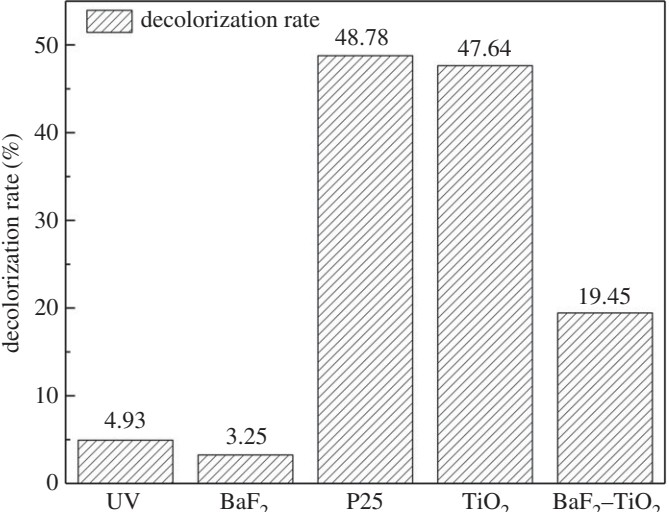

**Figure 8.** Effect of different catalysts on the decolorization rate of methyl orange solutions irradiated with UV light ($C_{BaF_2}$, $C_{TiO_2}$, $C_{P25}$: 1 g l$^{-1}$; $C_{0.35-BaF_2-TiO_2}$: 1.35 g l$^{-1}$).

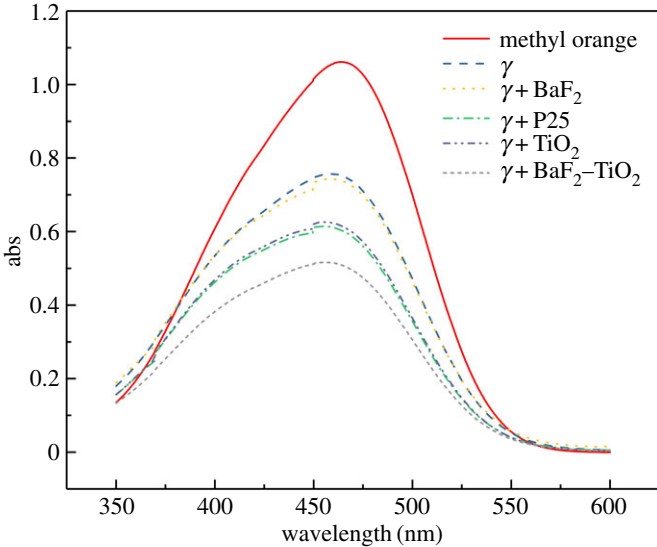

**Figure 9.** Ultraviolet–visible spectra before and after γ-ray irradiation of methyl orange solutions with different catalysts ($C_{BaF_2}$, $C_{TiO_2}$, $C_{P25}$: 1 g l$^{-1}$; $C_{0.75-BaF_2-TiO_2}$: 1.75 g l$^{-1}$).

sample. The theory is also supported by the catalytic ability of different composite catalysts in the experiment (the higher the BaF$_2$ content, the lower the catalytic effect).

### 3.2.3. Gamma-ray catalytic degradation of methyl orange

For these experiments, the γ-ray source was $^{60}$Co at a dose rate of 0.69 kGy h$^{-1}$ and the catalyst concentration was 1 g l$^{-1}$, which is identical to that used for the UV experiments. The UV–visible spectrum and the decolorization rate of the methyl orange solution after irradiation for 1 h are shown in figures 9 and 10, respectively. As seen, when γ-rays are used to irradiate the methyl orange solution, the decolorization rate of methyl orange is only 29.61%, indicating that γ-rays do indeed elicit some degradation of methyl orange. However, the decolorization rate of methyl orange hardly changed following the addition of BaF$_2$ as a catalyst. This result indicates that BaF$_2$ alone cannot be used to catalyse the degradation of methyl orange solution by γ-ray irradiation. In the presence of TiO$_2$ and P25, the decolorization rates of the methyl orange solution reached 42.21% and 43.34%, respectively, which clearly indicate that TiO$_2$ and P25 are capable of catalysing γ-ray degradation of methyl orange. However, since TiO$_2$ has a low utilization rate of high-energy rays, its catalytic effect is

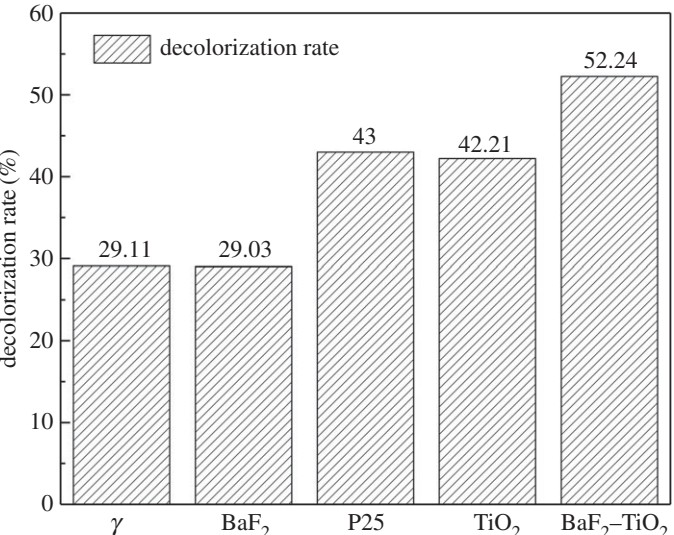

**Figure 10.** Effect of different catalysts on the decolorization rate of γ-ray irradiated methyl orange solutions ($C_{BaF_2}$, $C_{TiO_2}$, $C_{P25}$: 1 g l$^{-1}$; $C_{0.75-BaF_2-TiO_2}$: 1.75 g l$^{-1}$).

limited. When BaF$_2$–TiO$_2$ is used as a catalyst, the decolorization rate of methyl orange solution was significantly higher. For example, when 0.75-BaF$_2$–TiO$_2$ is used as a catalyst, the decolorization rate of methyl orange solution reached 52.24%, which is approximately 10% higher than that of TiO$_2$ alone. The possible reason is that BaF$_2$ is present in the composite catalyst, and BaF$_2$ as a detector material has a higher absorption cross-section for γ-rays than TiO$_2$. BaF$_2$ can effectively absorb high-energy radiation and emit ultraviolet light of 220 and 315 nm, which is then used to excite TiO$_2$ for photocatalysis. Therefore, the prepared composite catalyst material can be more effectively used for degrading methyl orange solutions with γ-radiation.

### 3.2.4. Electron beam catalytic degradation of methyl orange

Although our previous experiments have demonstrated that the prepared composite catalyst is weak to catalyse ultraviolet photodegradation of organic matter and good to catalyse γ-ray degradation of organic matter, whether the composite catalyst can be used to catalyse EB degradation of methyl orange solution needs further investigation.

For these experiments, the catalyst concentration was 1 g l$^{-1}$, the methyl orange solution concentration was 50 mg l$^{-1}$, and the sample was irradiated with a 10 MeV accelerator at a dose rate of 2 kGy r$^{-1}$. The ultraviolet–visible scanning spectrum and the methyl orange decolorization rate before and after irradiation are shown in figures 11 and 12, respectively. As before, BaF$_2$ demonstrates no catalytic ability for EB degradation of methyl orange, and the addition of P25 and the composite catalyst (0.75-BaF$_2$–TiO$_2$) only slightly enhances the EB degradation of methyl orange; its decolorization rate is 54.95% compared with a simple EB irradiation of methyl orange. Since the rate increased by only approximately 5%, it is clear that the prepared composite catalyst has almost no catalytic ability for EB degradation of organic matter. The possible reason for this is that the energy of the EB is too high, beyond the absorption range of BaF$_2$ and TiO$_2$. Therefore, the addition of the catalyst in this experiment has almost no effect. In general, the composite catalyst is most suitable for γ-irradiation degradation of methyl orange solution.

## 3.3. Effect of the BaF$_2$-to-TiO$_2$ doping ratio on methyl orange degradation

To determine the effect of BaF$_2$ and TiO$_2$ doping ratios on the degradation of methyl orange, the TiO$_2$ concentration in the methyl orange solution was maintained at 1 g l$^{-1}$ and the solution was irradiated by a $^{60}$Co source at a dose rate of 0.69 kGy h$^{-1}$ for 1 h. Of all the different BaF$_2$-to-TiO$_2$ doping ratios in the BaF$_2$–TiO$_2$ composites examined, the highest catalytic degradation of methyl orange solution was observed when $C_{BaF_2} : C_{TiO_2} = 0.75$. In this case, the decolorization rate reached 52.24% (figure 13). As the proportion of BaF$_2$ increased, the decolorization rate of methyl orange decreased slightly. This may be because an appropriate amount of BaF$_2$ can increase the catalytic performance of the

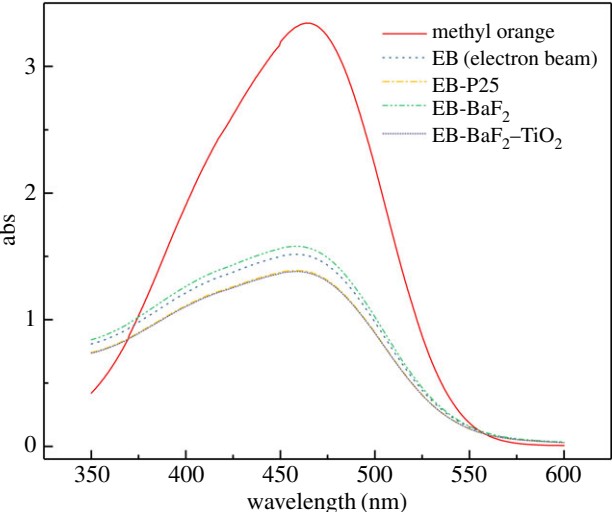

**Figure 11.** Ultraviolet–visible spectra before and after EB irradiation of methyl orange solutions with different catalysts ($C_{BaF_2}$, $C_{P25}$: 1 g l$^{-1}$; $C_{0.75\text{-}BaF_2\text{-}TiO_2}$: 1.75 g l$^{-1}$).

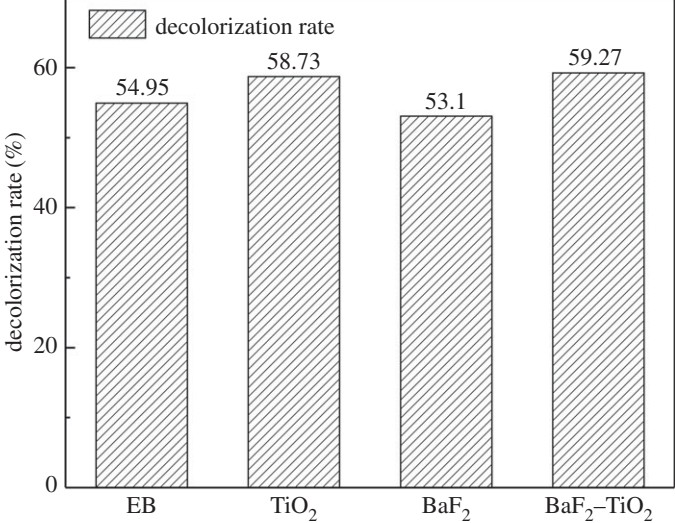

**Figure 12.** Effect of different catalysts on the decolorization rate of methyl orange solutions irradiated with an EB ($C_{BaF_2}$, $C_{P25}$: 1 g l$^{-1}$; $C_{0.75\text{-}BaF_2\text{-}TiO_2}$: 1.75 g l$^{-1}$).

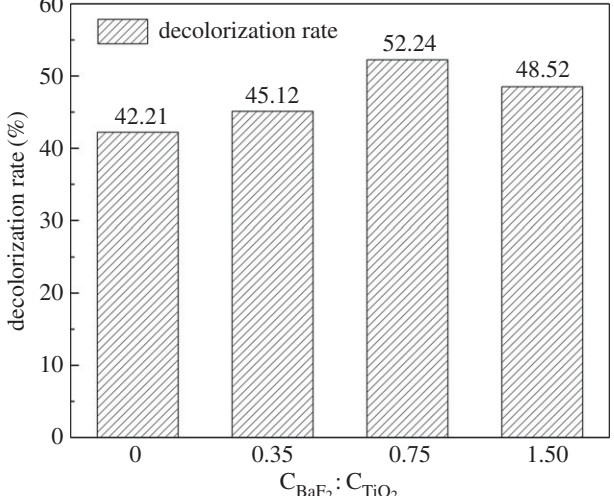

**Figure 13.** Effect of the BaF$_2$-to-TiO$_2$ doping ratio on the decolorization rate of methyl orange solutions.

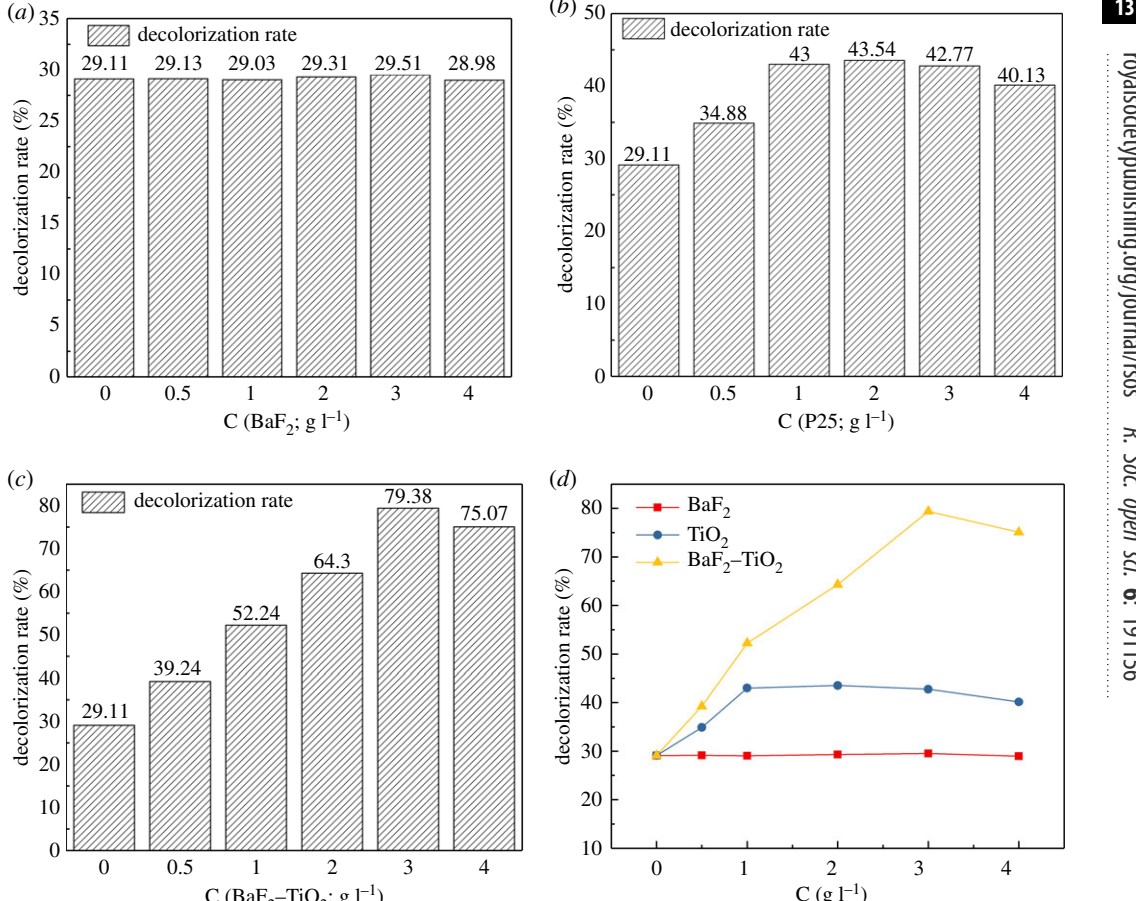

**Figure 14.** Effect of different catalyst concentrations on the decolorization rate of γ-irradiated methyl orange solutions ((a) BaF$_2$, (b) P25, (c) 0.75-BaF$_2$–TiO$_2$).

composite catalyst, while the excess BaF$_2$ blocks the TiO$_2$ surface. That is, the excess BaF$_2$ affects the formations of •OH and defects on the TiO$_2$ surface that would catalytically degrade methyl orange molecules. In addition, as the content of BaF$_2$ in the composite catalyst increases, the specific surface decreases, which may be one of the reasons for the decrease in the catalytic ability of 1.5-BaF$_2$–TiO$_2$. Therefore, we conclude that the optimum BaF$_2$-to-TiO$_2$ doping ratio is 0.75 ($C_{BaF_2} : C_{TiO_2} = 0.75$).

## 3.4. Effect of different catalyst concentrations on γ-irradiated methyl orange solutions

The above experiments have shown that the catalyst performs best when the composite catalyst is synthesized in a BaF$_2$ : TiO$_2$ ratio of 0.75 : 1. However, the effect of different catalyst concentrations on the decolorization rate of γ-irradiated methyl orange solution needs further investigation. In order to compare these effects fairly, the experiments of two series of P25 and BaF$_2$ were added under the same conditions. In this way, the superiority of 0.35-BaF$_2$–TiO$_2$ as a γ-irradiation catalyst was established.

To ascertain the effect of the catalyst concentration on the decolorization rate of methyl orange solution under γ-irradiation conditions, the decolorization rate of methyl orange after γ-irradiation for 1 h was determined. As seen from figure 14a, the decolorization rate of methyl orange is approximately 29% as the BaF$_2$ concentration is changed. Clearly, BaF$_2$ does not catalyse the degradation of methyl orange by γ-irradiation. At very high BaF$_2$ concentration, BaF$_2$ actually hinders the absorption of γ-rays by the solution and affects the degradation of methyl orange. The effect of P25 as a catalyst on the decolorization rate of γ-irradiated methyl orange solution is shown in figure 14b. As seen, the catalytic effect initially increases and then stabilizes as the P25 concentration is increased further. The decolorization rate is approximately 43% at a P25 concentration of 1 g l$^{-1}$; further increases in the P25 concentration do not significantly increase the catalytic effect. In fact, the catalytic effect actually decreases when the P25 concentration exceeds 3 g l$^{-1}$. These results may be understood in terms of the limited absorption of γ-rays by the solution at high P25 concentrations, which negatively affect the degradation effect. The effect

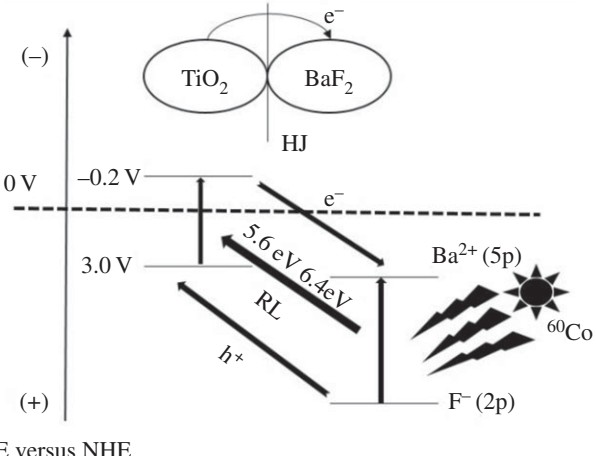

**Figure 15.** Scheme of the mechanism of BaF$_2$–TiO$_2$ γ-irradiation catalytic reaction.

of different mass concentration BaF$_2$–TiO$_2$ composites (0.75-BaF$_2$–TiO$_2$) on the decolorization rate of γ-irradiated methyl orange solution is shown in figure 14c. Although the trend in the decolorization rate is similar to that when P25 is used as the catalyst, the BaF$_2$–TiO$_2$ composite demonstrates a superior catalytic effect. When the TiO$_2$ concentration in the composite catalyst is 1 g l$^{-1}$, the decolorization rate of the methyl orange solution is 52.24%, and when its concentration reaches the optimum value of 3 g l$^{-1}$, there are more catalysts in the solution, which will produce more active particles, and the decolorization rate increased from 29.1 to 79.38%, compared with the same dose using only γ-irradiation. At this point, the catalyst concentration in the solution has reached saturation. A continued increase in concentration will hinder the absorption of γ-rays by the methyl orange solution, further affecting the yield of active particles produced by water radiolysis, and resulting in a decrease in catalytic effect.

## 3.5 Mechanism of composite catalyst

The mechanism of high-energy radiation used to induce TiO$_2$–BaF$_2$ composite catalyst γ-radiation catalysis is not yet clear. It is true that the composite catalyst can be excited by $^{60}$Co irradiation source, thus interior UV from RL by radio-sensitive BaF$_2$ should be a possible route. When the composite catalyst was irradiated with γ-irradiation, Ba$^{2+}$ (5p) was excited to Ba$^{2+}$ (5p*) leaving a hole, and then electrons from F$^-$ (2p) valence band to the cation Ba$^{2+}$ (5p) level with the release of 5.6 eV, 6.4 eV radiation (as shown in equations (3.1) and (3.2)) [17]. In addition, an interior electric field developed in BaF$_2$–TiO$_2$ depletion layer is another likely scheme. It is believed that the formed TiO$_2$ particles are probably combined with the BaF$_2$ surface via the Ti–O–Ba structural units. Since the energy band structures of TiO$_2$ and BaF$_2$ are different, a typical 'HJ' would be formed between TiO$_2$ and BaF$_2$. Due to their energy bands, TiO$_2$ and BaF$_2$ will bend into each other within this HJ that benefits charge separation within composite catalysts. An inner electronic field is thus established in the HJ directed from BaF$_2$ to TiO$_2$. Under this inner electric field, radiation-induced electrons in the TiO$_2$ will drift into the BaF$_2$ to endow the composite with irradiation catalytic activity (as shown in equation (3.3)) [21–23]. We believe that the γ-radiation catalytic mechanism of composite catalyst happens near BaF$_2$ and TiO$_2$ and seems to hybridize γ-irradiation and UV, as illustrated in figure 15.

$$Ba^{2+}(5p) + \gamma \rightarrow Ba^{2+}(5p*) \rightarrow (h^+)_{Ba^{2+}-(5p)}, \tag{3.1}$$

$$(e^-)_{F^-}(2p) \rightarrow (h^+)_{Ba^{2+}}(5p) \rightarrow 5.6, 6.4 \text{ eV radiation} \tag{3.2}$$

and

$$TiO_2(e^- + h^+) + BaF_2 \rightarrow TiO_2(h^+) + BaF_2(e^-). \tag{3.3}$$

# 4. Conclusion

In this study, BaF$_2$–TiO$_2$ composite catalysts were synthesized by a sol–gel method. The composites were characterized using XRD and SEM analyses, which showed that BaF$_2$ and TiO$_2$ were successfully synthesized and formed a PN-like structure (similar to a bridge) on the surface of the composite catalyst. For experiments on the degradation of methyl orange under UV, γ and EB radiation, the

composite catalyst showed better γ-radiation catalytic activity than under other conditions. The mechanism seems to be that $BaF_2$–$TiO_2$ composite can effectively absorb γ-rays to stimulate $BaF_2$ and emit ultraviolet light, which can then excite $TiO_2$ to generate photo-charge ($e^-$/$h^+$). In addition, photo-charge separation is enhanced by the HJ effect of the composite catalysts. Therefore, more active particles capable of degrading methyl orange are produced.

The optimum $BaF_2$-to-$TiO_2$ doping ratio in the composite material was determined as 0.75, and the catalyst Ti/Ba prepared according to this ratio is approximately 3. In the experiment of γ-irradiation degradation of methyl orange, the optimum concentration of $BaF_2$–$TiO_2$ composite was $3 \text{ g l}^{-1}$. At this concentration, the solution was irradiated for 1 h under the same conditions, and the decolorization rate of the methyl orange solution was increased from 29.1 to 79.38%, compared with the same dose using only γ-irradiation. Overall, the $BaF_2$–$TiO_2$ composite material prepared herein is an excellent γ-irradiation degradation methyl orange catalyst.

Data accessibility. Composite material characterization results and methyl orange solution degradation data are available from the Dryad Digital Repository: https://doi.org/10.5061/dryad.r9162p5 [38].

Authors' contributions. Y.L., G.W. and T.C.: substantial contributions to conception and design, or acquisition of data, or analysis and interpretation of data; G.W., X.F. and S.H.: drafting the article or revising it critically for important intellectual content; H.F., Q.S. and D.H.: final approval of the version to be published; W.J.: agreement to be accountable for all aspects of the work in ensuring that questions related to the accuracy or integrity of any part of the work are appropriately investigated and resolved.

Competing interests. We declare we have no competing interests.

Funding. This study was supported by the National Natural Science Foundation of China (grant nos. 11405086 to Y.L., G.W., S.H., Q.S., D.H. and W.J., 51878611 to T.C. and 51608480 to H.F.) and the Fundamental Research Funds for the Central Universities (grant no. NS2017037 to X.F.).

Acknowledgements. The authors thank PAPD (a project funded by the Priority Academic Program Development of Jiangsu Higher Education).

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
