## [Reviewer comments · Royal Society Open Science]

Review History

RSOS-191156.R0 (Original submission)

Review form: Reviewer 1

Is the manuscript scientifically sound in its present form?

No

Are the interpretations and conclusions justified by the results?

No

Is the language acceptable?

Yes

Do you have any ethical concerns with this paper?

No

Have you any concerns about statistical analyses in this paper?

No

Recommendation?

Major revision is needed (please make suggestions in comments)

Comments to the Author(s)

Major Revision

This work seems to report the effect of ultraviolet light, gamma ray and electron beam irradiation elevated degradation of methyl orange by catalyst. The aim is clear and meaningful. As a researcher in radiation chemistry, I know the author has tried hard work. However, I think the manuscript cannot be published in this level. The author should try to improve their work once more. If the authors can improve their work on following suggestions, I can recommend it.

Some suggestions can be as follow:

1. In page 4, in introduction part, the author seems describe the shortcoming for ionization irradiation. The author seems to solve this problem, whereas the description is in short. I think the author should try more to explain their innovation.
2. In page 5, in experiment part, a scheme is used. If possible, I think the author should use chemical equation. If the method is correct, more references should be cited.
3. In page 6-9, the characterization of catalyst is insufficient. The author just uses XRD to certify or ensure sample. I think that is in-suitable, the author should use more method to authenticate species. Additionally, for catalytic level evaluation, the specific surface area is very important. The author should give the datum in this field.
4. In catalytic degradation part, the author should give the efficiency of catalyst itself.
5. I don't know why the author use the catalyst with difference in weight. (BaF₂, TiO₂, P25: 1 g L⁻¹; mBaF₂-TiO₂: 1.75 g L⁻¹) ?
6. The author should give the effect of difference in specific surface area on catalysis efficiency.
7. Generally, the author give the result in catalysis efficiency. However, main mechanism is unknown. I think mechanism is more important, the author should try to explain it clearly.
8. The absorbed dose used seems low, why this dose used?
9. The atmosphere during irradiation is very important, the author should be care of this item.

Review form: Reviewer 2 (Vijaya Kumar V)

Is the manuscript scientifically sound in its present form?

No

Are the interpretations and conclusions justified by the results?

No

Is the language acceptable?

Yes

Do you have any ethical concerns with this paper?

No

Have you any concerns about statistical analyses in this paper?

No

Recommendation?

Major revision is needed (please make suggestions in comments)

Comments to the Author(s)

Please see attachment (Appendix A).

Review form: Reviewer 3**Is the manuscript scientifically sound in its present form?**

No

Are the interpretations and conclusions justified by the results?

Yes

Is the language acceptable?

Yes

Do you have any ethical concerns with this paper?

No

Have you any concerns about statistical analyses in this paper?

No

Recommendation?

Major revision is needed (please make suggestions in comments)

Comments to the Author(s)

The manuscript describes the synthesis, characterization and evaluation of a BAF2-TiO₂ nanocomposite as a catalyst at the irradiation degradation of methyl orange. It focuses a very interesting subject and the techniques were well used.

The weaknesses of the manuscript are related with some lack of experimental information and explanation, although discussion is ok, and with some presentation issues. Consequently, corrections are needed and authors should present the information in a more clear and precise way.

Points that should be clarified:

Page 3:

- Line 21: "Composites" instead of "composites"
- Line 23-24: Authors may not generalize the catalytic activity for organic matter since they have just tested methyl orange solutions. The sentence needs to be rewritten, something as " ... has a good catalytic effect on degradation of methyl orange by gamma irradiation".
- Line 30: a hyphen is missing
- Line 49: "Like traditional gamma-ray irradiation technology" at the beginning of the sentence does not make sense. Sentence may start at "However..."

Page 4:

Attention should be given to the 3.2 section.

- Line 30-31: "(5 mL concentrated hydrochloric acid)" is not needed and induce confusion.
- Line 42: Last sentence must include the explanation about the way authors refer to the pristine and obtained materials. For instance, it must be said here, and not just at page 9, that synthesized TiO₂ will be called "TiO₂" while "P25" will be used to designate the commercial TiO₂ obtained

from Aladdin Chemical Reagent – this will facilitate the reading and comprehension of the manuscript.

- Please check Figure 1, some letters are missing. Also its legend should be changed, ex: Schematic representation of the preparation...

Page 5:

- Please indicate the version of software program ImagePro Plus.

- Attention to section 3.4, please clarify: units of dose rate, why different dose rates and methyl orange concentrations were used at gamma irradiation and electron beam irradiations? This is important information that is not present, someone who is not familiar with the techniques might not understand it.

Page 6:

4.1 section

Authors state at 3.2 section that they have studied different BaF₂ and TiO₂ contents but just present results for one ratio. Something about the other ratios must be said (are they identical? The results shown represent all the ratios?).

Page 7 (and table from page 8):

- Lines 40-42: must be rewritten, it is not very clear

- Table is not clear also, according to the legend the columns should exchange position

Section 4.2 – 4.4:

- The difference in the composite concentrations tested must be explained (not just stated).

- Ratio mBaF₂: mTiO₂ should be clarified and also presented at the legends of figures 5, 7, 9 and 13.

- The value of decolorization rate obtained with BaF₂-TiO₂ at Figure 8 does not match any of those presented at Figure 11! Why?

- Should be mBaF₂: mTiO₂ instead of BaF₂: TiO₂ at Figure 11.

- At section 4.4 authors mentioned the effect of TiO₂ as catalyst but graphical information at Figure 12 b shows P25. Please correct.

Conclusions:

Authors may not generalize the catalytic activity for organic matter since they have just tested methyl orange solutions. Please correct.

Decision letter (RSOS-191156.R0)

30-Jul-2019

Dear Sir Wang:

Title: Irradiation-catalyzed degradation of methyl orange using BaF₂-TiO₂ nanocomposite catalysts prepared by a sol-gel method

Manuscript ID: RSOS-191156

The editor assigned to your manuscript has now received comments from reviewers. We would

like you to revise your paper in accordance with the referee and Subject Editor suggestions which can be found below (not including confidential reports to the Editor). Please note this decision does not guarantee eventual acceptance.

Please submit your revised paper before 22-Aug-2019. Please note that the revision deadline will expire at 00.00am on this date. If we do not hear from you within this time then it will be assumed that the paper has been withdrawn. In exceptional circumstances, extensions may be possible if agreed with the Editorial Office in advance. We do not allow multiple rounds of revision so we urge you to make every effort to fully address all of the comments at this stage. If deemed necessary by the Editors, your manuscript will be sent back to one or more of the original reviewers for assessment. If the original reviewers are not available we may invite new reviewers.

RSC Associate Editor:
Comments to the Author:
(There are no comments.)

RSC Subject Editor:
Comments to the Author:
(There are no comments.)

Reviewers' Comments to Author:

Reviewer: 1

Comments to the Author(s)

Major Revision

This work seems to report the effect of ultraviolet light, gamma ray and electron beam irradiation elevated degradation of methyl orange by catalyst. The aim is clear and meaningful. As a researcher in radiation chemistry, I know the author has tried hard work. However, I think the manuscript cannot be published in this level. The author should try to improve their work once more. If the authors can improve their work on following suggestions, I can recommend it.

Some suggestions can be as follow:

1. In page 4, in introduction part, the author seems describe the shortcoming for ionization irradiation. The author seems to solve this problem, whereas the description is in short. I think the author should try more to explain their innovation.
2. In page 5, in experiment part, a scheme is used. If possible, I think the author should use chemical equation. If the method is correct, more references should be cited.
3. In page 6-9, the characterization of catalyst is insufficient. The author just uses XRD to certify or ensure sample. I think that is in-suitable, the author should use more method to authenticate species. Additionally, for catalytic level evaluation, the specific surface area is very important. The author should give the datum in this field.
4. In catalytic degradation part, the author should give the efficiency of catalyst itself.
5. I don't know why the author use the catalyst with difference in weight. (BaF₂, TiO₂, P25: 1 g L⁻¹; mBaF₂-TiO₂: 1.75 g L⁻¹) ?
6. The author should give the effect of difference in specific surface area on catalysis efficiency.
7. Generally, the author give the result in catalysis efficiency. However, main mechanism is unknown. I think mechanism is more important, the author should try to explain it clearly.
8. The absorbed dose used seems low, why this dose used?
9. The atmosphere during irradiation is very important, the author should be care of this item.

Reviewer: 2

Comments to the Author(s)

Please see attachment.

Reviewer: 3

Comments to the Author(s)

The manuscript describes the synthesis, characterization and evaluation of a BAF₂-TiO₂ nanocomposite as a catalyst at the irradiation degradation of methyl orange. It focuses a very interesting subject and the techniques were well used.

The weaknesses of the manuscript are related with some lack of experimental information and explanation, although discussion is ok, and with some presentation issues. Consequently, corrections are needed and authors should present the information in a more clear and precise way.

Points that should be clarified:

Page 3:

- Line 21: "Composites" instead of "composites"
- Line 23-24: Authors may not generalize the catalytic activity for organic matter since they have just tested methyl orange solutions. The sentence needs to be rewritten, something as " ... has a good catalytic effect on degradation of methyl orange by gamma irradiation".

- Line 30: a hyphen is missing
- Line 49: "Like traditional gamma-ray irradiation technology" at the beginning of the sentence does not make sense. Sentence may start at "However..."

Page 4:

Attention should be given to the 3.2 section.

- Line 30-31: "(5 mL concentrated hydrochloric acid)" is not needed and induce confusion.
- Line 42: Last sentence must include the explanation about the way authors refer to the pristine and obtained materials. For instance, it must be said here, and not just at page 9, that synthesized TiO₂ will be called "TiO₂" while "P25" will be used to designate the commercial TiO₂ obtained from Aladdin Chemical Reagent - this will facilitate the reading and comprehension of the manuscript.
- Please check Figure 1, some letters are missing. Also its legend should be changed, ex: Schematic representation of the preparation...

Page 5:

- Please indicate the version of software program ImagePro Plus.
- Attention to section 3.4, please clarify: units of dose rate, why different dose rates and methyl orange concentrations were used at gamma irradiation and electron beam irradiations? This is important information that is not present, someone who are not familiar with the techniques might not understand it.

Page 6:

4.1 section

Authors state at 3.2 section that have studied different BaF₂ and TiO₂ contents but just present results for one ratio. Something about the other ratios must be said (are they identical? The results shown represent all the ratios?).

Page 7 (and table from page 8):

- Lines 40-42: must be rewritten, it is not very clear
- Table is not clear also, according to the legend the columns should exchange position

Section 4.2 - 4.4:

- The difference in the composite concentrations tested must be explained (not just stated).
- Ratio mBaF₂: mTiO₂ should be clarified and also presented at the legends of figures 5, 7, 9 and 13.
- The value of decolorization rate obtained with BaF₂-TiO₂ at Figure 8 does not match any of those presented at Figure 11! Why?
- Should be mBaF₂: mTiO₂ instead of BaF₂: TiO₂ at Figure 11.
- At section 4.4 authors mentioned the effect of TiO₂ as catalyst but graphical information at Figure 12 b shows P25. Please correct.

Conclusions:

Authors may not generalize the catalytic activity for organic matter since they have just tested methyl orange solutions. Please correct.

Author's Response to Decision Letter for (RSOS-191156.R0)

See Appendix B.

Decision letter (RSOS-191156.R1)

02-Sep-2019

Dear Sir Wang:

Title: Irradiation-catalyzed degradation of methyl orange using BaF₂-TiO₂ nanocomposite catalysts prepared by a sol-gel method
Manuscript ID: RSOS-191156.R1

It is a pleasure to accept your manuscript in its current form for publication in Royal Society Open Science. The chemistry content of Royal Society Open Science is published in collaboration with the Royal Society of Chemistry.

RSC Associate Editor
Comments to the Author:
(There are no comments.)

Reviewer(s)' Comments to Author:

Appendix A

Comments to author:

1. Why the authors choose methyl orange dye as model dye pollutant?
2. In page no-2, line no-14, 1. Summary must be change to Abstract.
3. The introduction part is not well-written. Also, there are few explanations of the rationale for the study design, line no-42 and 51. The novelty of the work was not highlighted in this manuscript?
4. In the heading 3.1 Reagents and Devices, what are the devices are used for the present study? The heading modify to chemicals and instruments to write elaborately the instruments utilized for present manuscript.
5. The heading 3.2 catalyst preparation was not clear? After the addition solution A and B to get $\text{BaF}_2\text{-TiO}_2$ composite and it was heated to 100°C for 2h followed by 400°C calcination. But why the author again heated to 105°C ? any specific reason is there.
6. I advised the author to modify $C_{\text{BaF}_2\text{-TiO}_2}$ to Conc. $\text{BaF}_2\text{-TiO}_2$ the following places in page-3, line-41; page-6, line-34 (Fig. 3); page-7, line-38 (Fig. 4); page-8, line-45 (Fig. 5); page-10, line-4 (Fig. 7); page-11, line-25 (Fig. 9); page-12, line-4 and 8; page-12, line-20 (Fig. 13).
7. In Fig.1 Dissolutio to modify Dissolution; BaF_2 -?
8. The author reported BaF_2 , TiO_2 and $\text{BaF}_2\text{-TiO}_2$ Nano size 38, 27 and 17 nm. But the nanoparticle size calculated by TEM analysis. The author mentioned one is average particle size.
9. From the Fig. 4C. fluoride peak missing?
10. The TiO_2 and $\text{BaF}_2\text{-TiO}_2$ composite material showed close decolourization ability for electron beam, but $\text{BaF}_2\text{-TiO}_2$ showed 10% higher decolourization efficiency why?
11. In some places the author mentioned UV scanning spectrum. The author modify UV-visible spectrum.
12. The author reported reusability study only second run, in general, at least repeat five times for reusability study and how the catalyst was reused write clearly?
13. The figures are not formatted as per the journal guidelines?
14. All the references to format as per the journal format? Examples; Ref-5 there was no issue and page numbers. Ref. 16 and 25 article end page was missing.
15. The authors reported decolourization rate increased 173% to 79.38% in some places, how 173% possible justify? or modify that one.
16. The authors can also report adsorption isotherms studies of methyl orange dye for $\text{BaF}_2\text{-TiO}_2$ catalyst to know the monolayer sorption capacity.
17. The authors are highly encouraged to seek an editing help from a native English speaking expertise.

Appendix B

Dear reviewers:

Manuscript ID: RSOS-191156

Title: Irradiation-catalyzed degradation of methyl orange using BaF₂-TiO₂ nanocomposite catalysts prepared by a sol-gel method

Author(s): Yongsheng Ling, Guang Wang, Ting Chen, Xionghui Fei, Song Hu, Qing Shan,

Daqian Hei, Huajun Feng, Wenbao Jia

With regard to the reviewers' comments and suggestions, we wish to reply as follows:

To reviewers 1

Firstly, I wish to clarify a question in the manuscript. In the originally submitted manuscript, the doping ratio of BaF₂ to TiO₂ was 0.5, 1 and 2. This was described based on the addition amount of the solution A and the solution B. However, according to the XRD and SEM Mapping characterization results, the actual doping ratio of the composite catalyst was 0.35, 0.75 and 1.5. So in the revised manuscript we corrected this problem.

Original manuscript: "Catalysts with different BaF₂ and TiO₂ contents (i.e., $m_{\text{BaF}_2}:m_{\text{TiO}_2} = 0.5, 1, 2$) were synthesized by changing the amount of solution B added to solution A.

Modified manuscript: "Catalysts with different BaF₂ and TiO₂ contents (i.e., $C_{\text{BaF}_2}:C_{\text{TiO}_2} = 0.35, 0.75, 1.5$) were synthesized by changing the amount of solution B added to solution A. And composite catalysts are named as X-BaF₂-TiO₂, where X (0.35, 0.75 and 1.5) corresponds to the weight ratio of BaF₂ to TiO₂ in the material."

1. In page 4, in introduction part, the author seems describe the shortcoming for ionization irradiation. The author seems to solve this problem, whereas the description is in short. I think the author should try more to explain their innovation.

.Response: Thank you very much. I have re-written this part, as following:

"In the series of processes, the yield of active particles plays an important role. However, like conventional ultraviolet radiation and ionizing radiation, there are problems requiring large doses and long reaction times before producing sufficient active particles [9-11]. To this end, a catalyst is added to increase the yield of the active particles during the irradiation, thereby reducing the irradiation time and dosage.

TiO₂ as a semiconductor material, when it is irradiated with UV, the active center of TiO₂ is photo-activated and an electron/hole (e^-/h^+) couples is obtained. And the electron/hole couples pair further react with oxygen and water to produce superoxide radical ion ($O_2^{\bullet-}$) and hydroxyl radical ($HO\bullet$); both of which are very reactive and strongly oxidizing to be capable of effectively catalyzing the degradation of organic pollutants and saving reaction time. Although TiO₂ has been demonstrated to be an effective catalyst in the presence of UV radiation, it is unsuitable for use as a catalyst in the presence of high energy and high permeability ionizing radiation because its band gap is only 3.2 eV. To compensate for the deficiency of traditional TiO₂ in high energy ionizing radiation catalytic oxidation, modifying TiO₂ through doping with certain materials that can utilize high energy radiation has been considered. As a scintillator material, BaF₂ is one of the activated materials that are used radioluminescent (RL) agents. Radioluminescence is the phenomenon where luminescence is produced in a material by the bombardment of γ -radiation or EB. The literature reveals that when BaF₂ is bombarded with high-energy ions, the electrons on the Ba²⁺(5p)

band are excited to the conduction band to leave the holes, and the electrons on the F⁻ (2p) valence band are transitioned to Ba(5p), which produces RL. Therefore, BaF₂ can effectively absorb high-energy radiation and emit ultraviolet light of 220 and 315 nm, which is then used to excite TiO₂ for photocatalysis to produce more active particles."

2. In page 5, in experiment part, a scheme is used. If possible, I think the author should use chemical equation. If the method is correct, more references should be cited.

Response: Thanks you for pointing this out. I added this part of the description and cite more literature, as following:

1. Pawar SG, Patil SL, Chougule MA, Jundale DM, Patil VB. 2011 Synthesis and characterization of nanocrystalline TiO₂ thin films. *Journal of Materials Science Materials in Electronics*. **22**, 260-264. (doi:10.1007/s10854-010-0125-8)
2. Tju H, Muzakki AT, Taufik A, Saleh R. 2017 Photo-, sono-, and sonophotocatalytic activity of metal oxide composites TiO₂/CeO₂ for degradation of dye. *American Institute of Physics Conference Series*; 2017. (doi:10.1063/1.4991138)
3. Wang W, Serp P, Kalck P, Faria JL. 2005 Photocatalytic degradation of phenol on MWNT and titania composite catalysts prepared by a modified sol-gel method. *Applied Catalysis B Environmental*. **56**, 305-312. (doi:10.1016/j.apcatb.2004.09.018)
4. Yu CT, Wang CF, Chen TY., Chang YT. 2008 Synthesis and characterization of radiation sensitive TiO₂/monazite photocatalyst. *Journal of Radioanalytical & Nuclear Chemistry*. **277**, 337-345. (doi:10.1007/s10967-007-7099-x)
5. Keihan AH, Hosseinzadeh R, Farhadian M, Kooshki H, Hosseinzadeh G. 2016 Solvothermal preparation of Ag nanoparticle and graphene co-loaded TiO₂ for the photocatalytic degradation of paraoxon pesticide under visible light irradiation. *RSC Advances*. **6**, 83673-83687. (doi:10.1039/C6RA19478H)
6. Llanos J, Brito I, Espinoza D, Sekar R., Manidurai P. 2018 A down-shifting Eu 3+ -doped Y₂WO₆/TiO₂ photoelectrode for improved light harvesting in dye-sensitized solar cells. *Royal Society Open Science*. **5**, 171054-171062. (doi:10.1098/rsos.171054)
7. Kormann, C., Bahnemann, DW, Hoffmann, MR. 1988 Preparation and characterization of quantum-size titanium dioxide. *J.phys.chem.* **92**, 5196-5201. (doi:10.1021/j100329a027)

8. Zhu, Y., Zhang, L, Gao C, Cao L. 2000 The synthesis of nanosized TiO₂ powder using a sol-gel method with TiCl₄ as a precursor. *Journal of Materials Science*. **35**, 4049-4054. (doi:10.1023/a:1004882120249)
9. Yanfei, L. 2007 Thermal stability of barium fluoride nano-powders with different particle size. *Chinese Journal of Materials Research*. **21**, 45-50. (doi:10.2514/1.26230)
10. Xiao-Hong L., Jian-Jun C., Chun-Xiao XU, Ming-Ming T. 2013 Preparation of BaF₂ and BaF₂:Ce(3+) Nanoparticles from W/O Microemulsion Systems. *Chinese Rare Earths*. **34**, 61-64. (This article does not have a DOI)

3. In page 6-9, the characterization of catalyst is insufficient. The author just uses XRD to certify or ensure sample. I think that is in-suitable, the author should use more method to authenticate species. Additionally, for catalytic level evaluation, the specific surface area is very important. The author should give the datum in this field.

Response: Thanks you for pointing this out. We have added some characterizations of the composite catalyst. The surface morphology of the catalyst and the distribution of Ti, Ba, O and F elements on the surface were determined by SEM. And the point scan of the composite catalyst combined with XRD results showed that TiO₂ and BaF₂ were successfully synthesized. In addition, we performed a BET analysis to determine the specific surface area of the composite catalyst. For the chapter of 4.1, I have re-written this part t. As following:

"4.1 Characterization of the catalyst

XRD patterns of the prepared BaF₂, TiO₂, and BaF₂-TiO₂ samples are shown in Figures 2, respectively. BaF₂ is of Frankdicksonite type composition; its crystal form is cubic, as indicated by consistency with the standard diffraction peaks in JCPDS card No. 01-001-0533. The (101), (112), (200), (105), (211), (204), (116), (220), and (215) crystal faces of TiO₂ are all anatase crystal forms, as indicated by consistency with the standard diffraction peaks in JCPDS card No. 01-071-1167. The BaF₂-TiO₂ composites have almost all 2θ values of BaF₂ and TiO₂, but the peak intensities are different. It is observed that the peaks belonging to BaF₂ gradually increased with the increase of BaF₂ content for the composite catalysts. In addition, according to Equation (1), the crystallite sizes of BaF₂ and TiO₂ are 38 nm and 27 nm.

Fig. 2 XRD patterns of (a) BaF₂, (b) TiO₂, (c) BaF₂-TiO₂ (0.35-BaF₂-TiO₂), (d) BaF₂-TiO₂ (0.75-BaF₂-TiO₂) and (e) BaF₂-TiO₂ (1.5-BaF₂-TiO₂)

Figure 2 shows the adsorption-desorption isotherm of the composite catalyst. It can be seen from Fig. 2 that all the samples have a typical type IV isotherm, indicating that the composite catalyst forms a mesoporous structure. From the results, the adsorption capacity of 0.75-BaF₂-TiO₂ and 0.35-BaF₂-TiO₂ is similar. In Table 1, the results from BET surface area measurements for the composite catalysts are given. As shown in the table 1, specific surface area and pore size of TiO₂ were 59.3 m²/g and 19.3 nm, while these values were 46.04-13.83 (m² g⁻¹) and 23.1-30.17 nm for the composite catalyst, changing with the initial BaF₂ to TiO₂ ratios from 0.35 to 1.5. Overall there is a decrease of the surface area when compared to that of the neat TiO₂. This indicated that nanophase TiO₂ particles were only embedded onto the surface of the BaF₂ substrates and introduction of TiO₂ onto the surface of BaF₂ results in certain reduction of specific surface area. The interface between these TiO₂ and BaF₂ should be the major reaction sites to catalytic reaction.

Fig. 3 N₂ adsorption-desorption isotherm and pore size distribution of composite catalysts

Table 1 Surface properties of Catalysts.

Catalyst	BET Surface Area ($\text{m}^2 \text{g}^{-1}$)	Pore Size (nm)
TiO_2	56	11.5
0.35- BaF_2 - TiO_2	46.04	23.1
0.75- BaF_2 - TiO_2	38.38	25.8
1.5- BaF_2 - TiO_2	13.83	30.17

The morphologies of BaF_2 , TiO_2 and composite catalysts were revealed by SEM investigation, and SEM images of BaF_2 , TiO_2 and representative composite catalyst (0.75- BaF_2 - TiO_2) shown in Figures 4a-d. As seen, the BaF_2 - TiO_2 composite particles are more regular and have a more defined shape than the prepared BaF_2 and TiO_2 , the TiO_2 and BaF_2 particles are layered. And the SEM image was processed using ImagePro Plus 6.0 software, which showed that the particle size of BaF_2 - TiO_2 composite is 175–200 nm, which is smaller than the 256–312 nm size of BaF_2 . Therefore, the TiO_2 particles hinder aggregation of the BaF_2 particles. In addition, due to the difference in band gap between TiO_2 and BaF_2 , the synthesized composite forms a PN-like structure (similar to a bridge) between BaF_2 and TiO_2 compared to pure TiO_2 crystal. The synthesized composite can induce specific electron transfer processes, improve charge separation efficiency, produce more active particles, and achieve better catalytic effect.

Fig. 4 SEM micrographs of (a) BaF₂, (b) TiO₂, and (c, d) 0.75-BaF₂-TiO₂

SEM mapping analyses were carried out to confirm the presence of TiO₂ and BaF₂ onto the surface of composite catalyst. Typically, the SEM spectral images of 0.75-BaF₂-TiO₂ is presented in Figures 5a-d, which are corresponding to the distribution of elements in the area of Figures 4d. Figures 5a-d expressly confirms the presence of Ti, O, Ba, F and the elemental distribution of Ti, O or Ba, F is essentially the same. Figure 5e shows the specific content of the surface of composite catalyst; the relative content of elemental Ti is 44.3% and that of Ba is 19.7%. In addition, the SEM image has indicated that the surface of the composite catalyst forms a composite structure consisting of two phases. In order to explore the composition of the two phases, two points (A and B) were selected in Figure 4d for SEM mapping analysis. The SEM mapping spectrum is presented in Figure 5f-g, the point A consists mainly of Ba and F elements, and the point B consists mainly of Ti and O elements. And the elemental composition of each point is almost the same as the elemental mass ratio of TiO₂ or BaF₂. Combined with the XRD patterns, the two phases that make up the composite catalyst are TiO₂ and BaF₂. As for other composite catalysts, their spectra are similar to those of 0.75-BaF₂-TiO₂, but the peak intensities are different due to different BaF₂ contents.

Fig. 5 SEM mapping of 0.75-BaF₂-TiO₂

(a: Ti elemental distribution; b: O elemental distribution; c: Ba elemental distribution; d: F elemental distribution; e: Elemental map of the catalyst surface; f: Elemental map of point A; g: Elemental map of point B)"

4. In catalytic degradation part, the author should give the efficiency of catalyst itself.

Response: Thank you very much. In Section 4.2 of the original manuscript, I explored the degradation efficiencies of BaF₂, TiO₂ and composite catalysts under different irradiation conditions. In addition, in the revised manuscript, I added a description of the methyl orange adsorption experiment with different catalyst. As following:

"The 1 h adsorption capacity of the different catalysts for methyl orange was tested in the dark using a 20 mg / L methyl orange solution as a solvent. The experimental results are shown in Fig. 6 (for the sake of clearer images, Figure 6 only shows the UV-visible spectrum of 0.75-BaF₂-TiO₂), P25 has the strongest adsorption capacity, and BaF₂ has almost no adsorption capacity as a catalyst. As for the composite catalyst, its adsorption capacity is between BaF₂ and TiO₂. Among them, 0.35-BaF₂-TiO₂ has the strongest adsorption capacity, and 1.5-BaF₂-TiO₂ has the lowest adsorption capacity. This is consistent with the N₂ adsorption results of Figure 3. In general, the adsorption of methyl orange solution by the catalyst was small within 1 h. However,

for the correctness of the data, the results of all experiments have removed the adsorption of methyl orange by the catalyst.

Fig. 6 Ultraviolet-visible spectra of methyl orange solution adsorption by the different catalysts

($C_{BaF_2}, C_{TiO_2}, C_{P25}: 1 \text{ g L}^{-1}; C_{0.75-BaF_2-TiO_2}: 1.75 \text{ g L}^{-1}$)

Since the concentration of the catalyst in the next catalytic experiment is based on the concentration of TiO_2 (i.e., the concentration of TiO_2 in the composite catalyst added the same as the concentration of pure TiO_2), it is necessary to know how much composite catalyst is required per gram of TiO_2 . Table 2 lists the total mass required when different composite catalysts contain 1g of TiO_2 ."

Table 2 Mass of composite catalyst of different composite catalysts required for 1 g of TiO_2 .

Catalyst	Ti/Ba	$m_{BaF_2-TiO_2}$ (g)
0.35-BaF ₂ -TiO ₂	6	1.35
0.75-BaF ₂ -TiO ₂	3	1.75
1.5-BaF ₂ -TiO ₂	1.5	2.5

5. I don't know why the author use the catalyst with difference in weight. (BaF₂, TiO₂, P25: 1 g L⁻¹; mBaF₂-TiO₂: 1.75 g L⁻¹) ?

Response: Thank you very much. I explained this problem in the revised manuscript, as following:

"Since the concentration of the catalyst in the next catalytic experiment is based on the concentration of TiO_2 (i.e., the concentration of TiO_2 in the composite catalyst added the same as the concentration of pure TiO_2), it is necessary to know how much composite catalyst is required per gram of TiO_2 . Table 2 lists the total mass required when different composite catalysts contain 1g of TiO_2 ."

Table 2 Mass of composite catalyst of different composite catalysts required for 1 g of TiO₂.

Catalyst	Ti/Ba	m _{BaF₂-TiO₂} (g)
0.35-BaF ₂ -TiO ₂	6	1.35
0.75-BaF ₂ -TiO ₂	3	1.75
1.5-BaF ₂ -TiO ₂	1.5	2.5

6. The author should give the effect of difference in specific surface area on catalysis efficiency.

Response: Thank you very much. The possible effects of specific surface area differences on catalytic efficiency are illustrated in the revised Section 4.3, as following:

"In addition, as the content of BaF₂ in the composite catalyst increases, the specific surface decreases, which may be one of the reasons for the decrease in the catalytic ability of 1.5-BaF₂-TiO₂."

7. Generally, the author give the result in catalysis efficiency. However, main mechanism is unknown. I think mechanism is more important, the author should try to explain it clearly.

Response: thank you very much for your suggestion. I described a possible mechanism in the revised manuscript, as following:

"The mechanism of high energy radiation used to induce TiO₂-BaF₂ composite catalyst γ -radiation catalysis is not yet clear. It is true that composite catalyst can be excited by ⁶⁰Co irradiation source, thus interior UV from radioluminescence(RL) by radio-sensitive BaF₂ should be a possible route. When the composite catalyst was irradiated with γ -irradiation, Ba²⁺(5p) was excited to Ba²⁺(5p*) leaving a hole, and then electrons from F(2p) valence band to the cation Ba²⁺(5p) level with the release of 5.6 eV, 6.4 eV radiation(As shown in Equation 11 and Equation 12)[36]. In addition, an interior electric field developed in BaF₂-TiO₂ depletion layer is another probably scheme. It is believed that the formed TiO₂ particles are probably combined with the BaF₂ surface via the Ti-O-Ba structural units .Since the energy band structures of TiO₂ and BaF₂ are different from each other, a typical "hetero-junction"(HJ) would be formed between TiO₂ and BaF₂. Due to the energy bands of TiO₂ and BaF₂ will bend into each other within this hetero-junction that benefit charge separation within composite catalysts. An inner electronic fields thus established in the hetero-junction directed from BaF₂ to TiO₂. Under this inner electric field, radiationinduced electrons in the TiO₂ will drift into the BaF₂ to endow the composite with irradiation -catalytic activity(As shown in Equation 13) [20-22]. We believe that the γ -radiation catalytic mechanism of composite catalyst happens by near BaF₂ and TiO₂ and seemed to hybridizing of γ -irradiation and UV as illustrated in Figure 15.

Fig. 15 Scheme of the mechanism of BaF₂-TiO₂ γ -irradiation catalytic reaction"

8. The absorbed dose used seems low, why this dose used?

Response: Thank you very much. My goal is to degrade methyl orange at a lower dose by using a composite catalyst, so we used a low dose. The description was revised in the introduction part.

As following:

"However, like conventional ultraviolet radiation and ionizing radiation, there are problems requiring large doses and long reaction times before producing sufficient active particles. To this end, the addition of a catalyst during irradiation has been proposed to increase the yield of active particles, thereby reducing the irradiation time and doses."

9. The atmosphere during irradiation is very important, the author should be care of this item.

Response: Thank you very much for your suggestion. In the follow-up work, I will study the effect of atmosphere on the catalytic effect. But due to time and some technical reasons, I can't complete this experiment now. I sincerely hope to get your understanding.

Special thanks to you for your good comments.

I tried my best to improve the manuscript and made some changes in the manuscript. These changes will not influence the content and framework of the paper.

I appreciate for editors and reviewers' warm work earnestly, and hope that the correction will meet with approval.

Once again, thank you very much for your comments and suggestions.

Thank you and best regards.

To reviewers 2

Firstly, I wish to clarify a question in the manuscript. In the originally submitted manuscript, the doping ratio of BaF₂ to TiO₂ was 0.5, 1 and 2. This was described based on the addition amount of the solution A and the solution B. However, according to the XRD and SEM Mapping characterization results, the actual doping ratio of the composite catalyst was 0.35, 0.75 and 1.5.

So in the revised manuscript we corrected this problem.

Original manuscript: "Catalysts with different BaF₂ and TiO₂ contents (i.e., $m_{\text{BaF}_2}:m_{\text{TiO}_2} = 0.5, 1, 2$) were synthesized by changing the amount of solution B added to solution A."

Modified manuscript: "Catalysts with different BaF₂ and TiO₂ contents (i.e., $C_{\text{BaF}_2}:C_{\text{TiO}_2} = 0.35, 0.75, 1.5$) were synthesized by changing the amount of solution B added to solution A. And composite catalysts are named as X-BaF₂-TiO₂, where X (0.35, 0.75 and 1.5) corresponds to the weight ratio of BaF₂ to TiO₂ in the material."

1. Why the authors choose methyl orange dye as model dye pollutant?

Response: Thank you very much. The reason for using methyl orange as the target pollutant is to study a method for degrading azo wastewater, and methyl orange is very representative. In

addition, in the manuscript, I also made a similar description. As following:

"With ongoing the textile printing and dyeing industry industrial development, a large amount of toxic and not easily degradable wastewater is continually being discharged into the environment. And the textile printing and dyeing industry mainly produces organic wastewater, with azo dyes as a typical pollutant [1, 2]. Therefore, the proper treatment of wastewater containing azo dyes(e.g., methyl orange) is currently a central area of research in water treatment."

2. In page no-2, line no-14, 1. Summary must be change to Abstract.

Response: Thank you very much. Done as suggested.

3. The introduction part is not well-written. Also, there are few explanations of the rationale for the study design, line no-42 and 51. The novelty of the work was not highlighted in this manuscript?

Response: Thank you very much for your suggestion. I have written the introduction section. In the revised manuscript, a more detailed explanation of the purpose and principle of the experiment, as following:

"With ongoing the textile printing and dyeing industry industrial development, a large amount of toxic and not easily degradable wastewater is continually being discharged into the environment. And the textile printing and dyeing industry mainly produces organic wastewater, with azo dyes as a typical pollutant. Therefore, the proper treatment of wastewater containing azo dyes(e.g., methyl orange) is currently a central area of research in water treatment.

In the field of wastewater treatment, UV degradation and ionizing radiation degradation are promising methods. These two methods have great potential for conversion of photon energy into chemical energy which can degrade the pollutants in water. For example, Studies have shown that a series of highly reactive particles (e.g., •OH, •H, and e⁻ aq) are produced after exposing water to high-energy radiation(As shown in Equation 1). Then these particles can react with aqueous pollutants by way of several reactions (i.e., addition, substitution, electron transfer, and bond cleavage) for the purpose of pollutant removal and water purification(As shown in Equation 2 and Equation 3).

The values in parentheses in Equation 1 are the radiochemical yield G values ($\mu\text{mol J}^{-1}$) of each of the active particles generated at a pH of 6-8. In Equation 2 and Equation 3, R represents organic pollutants, and $R\cdot$ represents intermediate products.

In this series of processes, the yield of active particles has the most important role. However, like conventional ultraviolet radiation and ionizing radiation, there are problems requiring large doses and long reaction times before producing sufficient active particles. To this end, We propose to add a catalyst during irradiation to increase the yield of active particles, thereby reducing the irradiation time and doses.

TiO_2 as a semiconductor material, when it is irradiated with UV, the active center of TiO_2 is photo-activated and an electron/hole (e^-/h^+) couples is obtained. And the electron/hole couples pair further react with oxygen and water to produce superoxide radical ion ($\text{O}_2^{\bullet-}$) and hydroxyl radical ($\text{HO}\cdot$); both of which are very reactive and strongly oxidizing to be capable of effectively catalyze the degradation of organic pollutants and save reaction time. Although TiO_2 has been demonstrated to be an effective catalyst in the presence of UV radiation, it is unsuitable for use as a catalyst in the presence of high energy and high permeability ionizing radiation because its band gap is only 3.2 eV. To compensate for the deficiency of traditional TiO_2 in high energy ionizing radiation catalytic oxidation, modifying TiO_2 through doping with certain materials that can utilize high energy radiation has been considered. As a scintillator material, BaF_2 is one of the activated materials that are used radioluminescent (RL) agents. Radioluminescence is the phenomenon where luminescence is produced in a material by the bombardment of γ -radiation or EB. The literature reveals that when BaF_2 is bombarded with high-energy ions, the electrons on the $\text{Ba}^{2+}(5p)$ band are excited to the conduction band to leave the holes, and the electrons on the F^-

(2p) valence band are transitioned to Ba(5p), which produces RL. Therefore, BaF₂ can effectively absorb high-energy radiation and emit ultraviolet light of 220 and 315 nm, which is then used to excite TiO₂ for photocatalysis to produce more active particles.

In this study, BaF₂-TiO₂ composites were prepared by sol gelation and characterized by XRD, SEM, and BET. The results show that the composite material forms an interesting Ti-F-Ba boundary microstructure, which established similar p-n junction potential between BaF₂-TiO₂. And the SEM mapping of the surface of the material reveals the elemental distribution of Ba, Ti, O and F on the surface of the composite prepared by different TiO₂-BaF₂ doping ratios. In addition, Methyl orange solution was irradiated by UV radiation, γ -ray radiation, and electron beams, and the catalytic activity of the composite for the degradation of methyl orange by these three types of radiation was studied. A possible mechanism of hybrid of RL and heterojunction (HJ) is proposed to illustrate this radiation catalytic behavior."

4. In the heading 3.1 Reagents and Devices, what are the devices are used for the present study?

The heading modify to chemicals and instruments to write elaborately the instruments utilized for present manuscript.

Response: Thank you very much. Done as suggested. As following:

"3.1 Chemicals and Instruments

Reagent raw materials: TiCl₄ ($\geq 99.5\%$, Sinopharm Chemical Reagent), HCl (36–38%, Sinopharm Chemical Reagent), citric acid ($\geq 99.5\%$, Nanjing Chemical Reagent), ammonia (25–28%, Nanjing Chemical Reagent), BaCl₂ ($\geq 99.5\%$, Nanjing Chemical Reagent), NaF ($\geq 98\%$, Nanjing Chemical Reagent), ethylenediaminetetraacetic acid (EDTA, $\geq 99.5\%$, Nanjing Chemical Reagent), quartz wool (1–3 μm , Nanjing Chemical Reagent), TiO₂ (P25, Aladdin Chemical Reagent), and methyl

orange (AR, Nanjing Chemical Reagent); Analytical Balances(A1004B, Yoke instrument), Muffle furnace(SLR-1200, Shanghai Daheng Optics), High temperature blast drying oven(XCT-0AS, Guangzhou Kenton), Magnetic stirrer(H01-1G, Shanghai Mei Yingpu)."

5. The heading 3.2 catalyst preparation was not clear? After the addition solution A and B to get $\text{BaF}_2\text{-TiO}_2$ composite and it was heated to 100°C for 2h followed by 400°C calcination. But why the author again heated to 105°C ? any specific reason is there.

Response: Thank you very much. Heating the composite again to 105°C is just to dry the material.

Because we have washed the material before.

"The obtained powder was washed with distilled water until no Cl^- was detected. Finally, the powder was dried in an oven at 105°C to obtain the $\text{BaF}_2\text{-TiO}_2$ material."

6. I advised the author to modify $\text{CBaF}_2\text{-TiO}_2$ to Conc. $\text{BaF}_2\text{-TiO}_2$ the following places in page-3, line-41; page-6, line-34 (Fig. 3); page-7, line-38 (Fig. 4); page-8, line-45 (Fig. 5); page-10, line-4 (Fig. 7); page-11, line-25 (Fig. 9); page-12, line-4 and 8; page-12, line-20 (Fig. 13).

Response: Thanks you for pointing this out. Inspired by you, I renamed the composite catalyst.

Now I can express them very clearly. As following:

"And composite catalysts are named as $\text{X-BaF}_2\text{-TiO}_2$, where X (0.35, 0.75 and 1.5) corresponds to the weight ratio of BaF_2 to TiO_2 in the material."

7. In Fig.1 Dissolutio to modify Dissolution; BaF_2 -?

Response: Thanks you for pointing this out. I have corrected it. As following:

8. The author reported BaF₂, TiO₂ and BaF₂-TiO₂ Nano size 38, 27 and 17 nm. But the nanoparticle size calculated by TEM analysis. The author mentioned one is average particle size.

Response: Thanks you for pointing this out. The Nano size of these catalysts was calculated by Scherrer's equation. But after your reminder, I found that this equation does not apply to all composite catalysts, so I removed the following part of the manuscript.

"The crystallite size of the TiO₂ and BaF₂ was calculated using Scherrer's equation (9)[33, 34]:

$$D = K\lambda / (\beta \cos\theta) \quad (9)$$

In addition, according to Equation (1), the crystallite sizes of BaF₂ and TiO₂ are 38 nm and 27 nm."

9. From the Fig. 4C. fluoride peak missing?

Response: Thank you very much. Following your guidance, I further characterized the composite catalyst. The distribution of the elements is described in the revised manuscript. As following:

"SEM mapping analyses were carried out to confirm the presence of TiO₂ and BaF₂ onto the surface of composite catalyst. Typically, the SEM spectral images of 0.75-BaF₂-TiO₂ is presented in Figures 5a-d, which are corresponding to the distribution of elements in the area of Figures 4d. Figures 5a-d expressly confirms the presence of Ti, O, Ba, F and the elemental distribution of Ti, O or Ba, F is essentially the same. Figure 5e shows the specific content of the surface of composite catalyst; the relative content of elemental Ti is 44.3% and that of Ba is 19.7%. In

addition, the SEM image has indicated that the surface of the composite catalyst forms a composite structure consisting of two phases. In order to explore the composition of the two phases, two points (A and B) were selected in Figure 4d for SEM mapping analysis. The SEM mapping spectrum is presented in Figure 5f-g, the point A consists mainly of Ba and F elements, and the point B consists mainly of Ti and O elements. And the elemental composition of each point is almost the same as the elemental mass ratio of TiO_2 or BaF_2 . Combined with the XRD patterns, the two phases that make up the composite catalyst are TiO_2 and BaF_2 . As for other composite catalysts, their spectra are similar to those of 0.75- BaF_2 - TiO_2 , but the peak intensities are different due to different BaF_2 contents.

Fig. 5 SEM mapping of $0.75\text{-BaF}_2\text{-TiO}_2$

(a: Ti elemental distribution; b: O elemental distribution; c: Ba elemental distribution; d: F elemental distribution; e: Elemental map of the catalyst surface; f: Elemental map of point A; g: Elemental map of point B)"

10. The TiO_2 and $\text{BaF}_2\text{-TiO}_2$ composite material showed close decolourization ability for electron beam, but $\text{BaF}_2\text{-TiO}_2$ showed 10% higher decolourization efficiency why?

Response: Thank you very much. In response to this question, I have made some explanations in the revised manuscript , as following:

"The possible reason is that BaF_2 is present in the composite catalyst, and BaF_2 as a detector material has a higher absorption cross section for γ -rays than TiO_2 . BaF_2 can effectively absorb high-energy radiation and emit ultraviolet light of 220 and 315 nm, which is then used to excite TiO_2 for photocatalysis. therefore, the prepared composite catalyst material can be more effectively used for degrading methyl orange solutions with γ -radiation."

11. In some places the author mentioned UV scanning spectrum. The author modify UV-visible spectrum.

Response: Thank you very much. Done as suggested.

12. The author reported reusability study only second run, in general, at least repeat five times for reusability study and how the catalyst was reused write clearly?

Response: Thanks you for pointing this out. Since the composite catalyst was attached to the quartz wool in the experiment, only a part of the composite catalyst could be recovered. So I didn't have enough catalyst to complete five experiments. I am very sorry that I can't answer your question. I have removed this part of the manuscript.

13. The figures are not formatted as per the journal guidelines?

Response: Thank you very much. In the revised manuscript, I have modified the figure format according to the articles published in the journal. I hope that as you think.

14. All the references to format as per the journal format? Examples; Ref-5 there was no issue and page numbers. Ref. 16 and 25 article end page was missing.

Response: Thanks you for pointing this out. I have modified all the journal formats. As following:

1. SU, Pukdee-Asa, Massakul, Ratanatamskul, Chavalit LU, MingChun. 2011 Effect of operating parameters on decolorization and COD removal of three reactive dyes by Fenton's reagent using fluidized-bed reactor. *Desalination*. **278**, 211-218. (doi:10.1016/j.desal.2011.05.022)
2. Vikrant K, Giri BS, Raza N, Roy K, Kim KH, Rai BN. 2018 Review: Recent advancements in bioremediation of dye: Current status and challenges. *Bioresource Technology*. **253**, 355-367. (doi:10.1016/j.biortech.2018.01.029)
3. You-Peng C, Shao-Yang L, Han-Qing Y, Hao Y, Qian-Rong L. 2008 Radiation-induced degradation of methyl orange in aqueous solutions. *Chemosphere*. **72**, 0-536. (doi:10.1016/j.chemosphere.2008.03.054)
4. Şolpan D, Güven O. 2002 Decoloration and degradation of some textile dyes by gamma irradiation. *Radiation Physics & Chemistry*. **65**, 549-558. (doi:10.1016/S0969-806X(02)00366-3)
5. Spinks JWT, Woods RJ. 1964 An introduction to radiation chemistry. *Mutation Research*. **658(1-2)**, 127-135. (doi:10.1016/j.mrrev.2007.10.004:)
6. Shah NS, Khan JA, Nawaz S, Khan HM. 2014 Role of aqueous electron and hydroxyl radical in the removal of endosulfan from aqueous solution using gamma irradiation. *Journal of Hazardous Materials*. **278**, 40-48. (doi:10.1016/j.jhazmat.2014.05.073)
7. Peng C, Yang D, An F, Li W, Li S, Ying N, Zhou L, Li Y, Wang C, Li S. 2015 Degradation of ochratoxin A in aqueous solutions by electron beam irradiation. *Journal of Radioanalytical & Nuclear Chemistry*. **306**, 39-46. (doi:10.1007/s10967-015-4086-5)
8. Kwon M, Yoon Y, Cho E, Jung Y, Lee BC, Paeng KJ, Kang JW. 2012 Removal of iopromide and degradation characteristics in electron beam irradiation process. *Journal of Hazardous Materials*. **227-228**, 126-134. (doi:10.1016/j.jhazmat.2012.05.022)
9. Wang J, Chu L. 2017 Research progress of ionizing irradiation technology on wastewater treatment. *Chinese Journal of Environmental Engineering*. **11**, 653-672. (doi:10.12030/j.cjee.201611148)
10. Sánchez-Polo M, López-Peñalver J, Prados-Joya G, Ferro-García MA, Rivera-Utrilla J. 2009 Gamma irradiation of pharmaceutical compounds, nitroimidazoles, as a new alternative for water treatment. *Water Research*. **43**, 4028-4036. (doi:10.1016/j.watres.2009.05.033)
11. Zheng BG., Zheng Z, Zhang JB., Luo XZ, Wang JQ. 2011 Degradation of the emerging contaminant ibuprofen in aqueous solution by gamma irradiation. *Desalination*. **276**, 379-385. (doi:10.1016/j.desal.2011.03.078)
12. Tang WZ, Zhang Z, An H, Quintana MO, Torres DF 1997 TiO₂/UV Photodegradation of Azo Dyes in Aqueous Solutions. *Environmental Technology Letters*. **18**, 1-12. (doi:10.1080/09593330.1997.9618466)
13. Jing WW, Li DQ, Li J, Li X F, Wu ZH, Liu YL. 2018 Photodegradation of dimethyl phthalate (DMP) by UV–TiO₂ in aqueous solution: operational parameters and kinetic analysis. *International Journal of Environmental Science & Technology*. **15**, 969-976. (doi:10.1007/s13762-017-1471-3)
14. Sampaio MJ, Pastrana-Martínez LM, Silva AM, Buijnsters JG, Han C, Silva CG, Carabineiro, S A, Dionysiou, D D, Faria, J L. 2015 Nanodiamond–TiO₂ composites for photocatalytic degradation of microcystin-LA in aqueous solutions under simulated solar light. *RSC Advances*. **5**, 58363-58370. (doi:10.1039/C5RA08812G)
15. Hu J, Cao Y, Wang K, Jia D. 2017 Green solid-state synthesis and photocatalytic hydrogen production activity of anatase TiO₂ nanoplates with super heat-stability. *RSC Advances*. **7**, 11827-11833. (doi: 10.1039/c6ra27160j)

16. Su P, Fu W, Yao H, Li L, Dong D, Fei F, Shuang F, Xue Y, Liu X, Yang H. 2017 Enhanced photovoltaic properties of perovskite solar cells by TiO₂ homogeneous hybrid structure. *Royal Society Open Science*. **4**, 170942-170948. (doi:10.1098/rsos.170942)
17. Yang J, Xia S, Wang K, Shi C. 1991 Electronic structure and luminescence of BaF₂ crystal. *Chinese Journal of Computation Physics*. **8**, 131-136. (This article does not have a DOI)
18. Wang CF, Yu CT, Lin BH, Lee JH. 2006 Synthesis and characterization of TiO₂/BaF₂/ceramic radio-sensitive photocatalyst. *Journal of Photochemistry & Photobiology A Chemistry*. **182**, 93-98. (doi:10.1016/j.jphotochem.2006.01.020)
19. Drozdowski W, Wojtowicz AJ. 2002 Fast 20 ns 5d–4f Luminescence and Radiation Trapping in BaF₂: Ce. *Nuclear Inst & Methods in Physics Research A*. **486**, 412-416. (doi:10.1016/S0168-9002(02)00744-1)
20. Arimoto O, Watanabe M, Tsujibayashi T, Azuma J, Kamada M, Nakanishi S, Itoh H, Itoh M. 2010 Photostimulated detection of radiation defects produced by UV light in BaF₂. *Radiation Measurements*. **45**, 356-358. (doi:10.1016/j.radmeas.2009.11.002)
21. Sah CT, Noyce RN, Shockley W. 1957 Carrier Generation and Recombination in P-N Junctions and P-N Junction Characteristics. *Proceedings of the Ire*. **45**, 1228-1243. (doi:10.1109/JRPROC.1957.278528)
22. Koizumi S, Watanabe K, Hasegawa M, Kanda, H. 2001 Ultraviolet emission from a diamond pn junction. *Science*. **292**, 1899-1901. (doi:10.1126/science.1060258)
23. Liu W, Chen SF. 2010 Visible-Light Activity Evaluation of p-n Junction Photocatalyst NiO/TiO₂ Prepared by Sol-Gel Method. *Advanced Materials Research*. **152-153**, 441-449. (doi:10.4028/www.scientific.net/AMR.152-153.441)
24. Pawar SG, Patil SL, Chougule MA, Jundale DM, Patil VB. 2011 Synthesis and characterization of nanocrystalline TiO₂ thin films. *Journal of Materials Science Materials in Electronics*. **22**, 260-264. (doi:10.1007/s10854-010-0125-8)
25. Tju H, Muzakki AT, Taufik A, Saleh R. 2017 Photo-, sono-, and sonophotocatalytic activity of metal oxide composites TiO₂/CeO₂ for degradation of dye. *American Institute of Physics Conference Series*; 2017. (doi:10.1063/1.4991138)
26. Wang W, Serp P, Kalck P, Faria JL. 2005 Photocatalytic degradation of phenol on MWNT and titania composite catalysts prepared by a modified sol-gel method. *Applied Catalysis B Environmental*. **56**, 305-312. (doi:10.1016/j.apcatb.2004.09.018)
27. Yu CT, Wang CF, Chen TY., Chang YT. 2008 Synthesis and characterization of radiation sensitive TiO₂/monazite photocatalyst. *Journal of Radioanalytical & Nuclear Chemistry*. **277**, 337-345. (doi:10.1007/s10967-007-7099-x)
28. Keihan AH, Hosseinzadeh R, Farhadian M, Kooshki H, Hosseinzadeh G. 2016 Solvothermal preparation of Ag nanoparticle and graphene co-loaded TiO₂ for the photocatalytic degradation of paraoxon pesticide under visible light irradiation. *RSC Advances*. **6**, 83673-83687. (doi:10.1039/C6RA19478H)
29. Llanos J, Brito I, Espinoza D, Sekar R., Manidurai P. 2018 A down-shifting Eu 3+ -doped Y₂WO₆/TiO₂ photoelectrode for improved light harvesting in dye-sensitized solar cells. *Royal Society Open Science*. **5**, 171054-171062. (doi:10.1098/rsos.171054)
30. Kormann, C., Bahnemann, DW, Hoffmann, MR. 1988 Preparation and characterization of quantum-size titanium dioxide. *J.phys.chem*. **92**, 5196-5201. (doi:10.1021/j100329a027)
31. Zhu, Y., Zhang, L, Gao C, Cao L. 2000 The synthesis of nanosized TiO₂ powder using a sol-gel method with TiCl₄ as a precursor. *Journal of Materials Science*. **35**, 4049-4054. (doi:10.1023/a:1004882120249)
32. Yanfei, L. 2007 Thermal stability of barium fluoride nano-powders with different particle size. *Chinese Journal of Materials Research*. **21**, 45-50. (doi:10.2514/1.26230)
33. Xiao-Hong L, Jian-Jun C, Chun-Xiao XU, Ming-Ming T. 2013 Preparation of BaF₂ and BaF₂:Ce(3+) Nanoparticles from W/O Microemulsion Systems. *Chinese Rare Earths*. **34**, 61-64. (This article does not have a DOI)

34. Burton AW, Ong K, Rea T, Chan IY. 2009 On the estimation of average crystallite size of zeolites from the Scherrer equation: A critical evaluation of its application to zeolites with one-dimensional pore systems. *Microporous & Mesoporous Materials*. **117**, 75-90. (doi:10.1016/j.micromeso.2008.06.010)
35. Chen R, Li Y, Yi Z, Li S, Xiang R, Xu N, Fang L. 2016 Effect of inorganic acid on the phase transformation of alumina. *Journal of Alloys & Compounds*. **699**, 170-175. (doi:10.1016/j.jallcom.2016.12.390)
36. Guojun KE, Zhang L, Xie Y. 2017 Measurement of specific surface area of aggregate based on Image-Pro Plus. *Concrete*. **9**, 157-160. (doi:10.3969/j.issn.1002-3550.2017.09.041)
- Magee JL. 1961 Radiation Chemistry. *Physical Chemistry*. **3**, 171-192. (doi:10.1146/annurev.ns.03.120153.001131)

15. The authors reported decolourization rate increased 173% to 79.38% in some places, how 173% possible justify? or modify that one.

Response: Thank you very much. Done as suggested.

"At this concentration, the solution was irradiated for 1 h under the same conditions, and the decolorization rate of the methyl orange solution was increased from 29.1% to 79.38%, compared with the same dose using only γ -irradiation."

16. The authors can also report adsorption isotherms studies of methyl orange dye for BaF₂-TiO₂ catalyst to know the monolayer sorption capacity.

Response: Thank you very much. Done as suggested.

"Catalyst adsorption of methyl orange

The 1 h adsorption capacity of the different catalysts for methyl orange was tested in the dark using a 20 mg / L methyl orange solution as a solvent. The experimental results are shown in Fig. 6 (for the sake of clearer images, Figure 6 only shows the UV-visible spectrum of 0.75-BaF₂-TiO₂), P25 has the strongest adsorption capacity, and BaF₂ has almost no adsorption capacity as a catalyst. As for the composite catalyst, its adsorption capacity is between BaF₂ and TiO₂. Among them, 0.35-BaF₂-TiO₂ has the strongest adsorption capacity, and 1.5-BaF₂-TiO₂ has the lowest adsorption capacity. This is consistent with the N₂ adsorption results of Figure 3. In general, the adsorption of methyl orange solution by the catalyst was small within 1 h. However,

for the correctness of the data, the results of all experiments have deduced the adsorption of methyl orange by the catalyst.

Fig. 6 Ultraviolet-visible spectra of methyl orange solution adsorption by the different catalysts ($C_{BaF_2}, C_{TiO_2}, C_{P25}: 1 \text{ g L}^{-1}; C_{0.75-BaF_2-TiO_2}: 1.75 \text{ g L}^{-1}$)"

17. The authors are highly encouraged to seek an editing help from a native Englishspeaking expertise.

Response: Thank you very much. I am sorry to bring you a bad reading experience. But before submitting the article, I have already sought an editing help from a native Englishspeaking expertise. In the revised manuscript, I have done my best to modify the grammar. I hope to get your understanding.

Special thanks to you for your good comments.

I tried my best to improve the manuscript and made some changes in the manuscript. These changes will not influence the content and framework of the paper.

I appreciate for editors and reviewers' warm work earnestly, and hope that the correction will meet with approval.

Once again, thank you very much for your comments and suggestions.

Thank you and best regards.

To reviewers 3

Firstly, I wish to clarify a question in the manuscript. In the originally submitted manuscript, the doping ratio of BaF₂ to TiO₂ was 0.5, 1 and 2. This was described based on the addition amount of the solution A and the solution B. However, according to the XRD and SEM Mapping characterization results, the actual doping ratio of the composite catalyst was 0.35, 0.75 and 1.5.

So in the revised manuscript we corrected this problem.

Original manuscript: "Catalysts with different BaF₂ and TiO₂ contents (i.e., $m_{\text{BaF}_2}:m_{\text{TiO}_2} = 0.5, 1, 2$) were synthesized by changing the amount of solution B added to solution A.

Modified manuscript: " Catalysts with different BaF₂ and TiO₂ contents (i.e., $C_{\text{BaF}_2}:C_{\text{TiO}_2} = 0.35, 0.75, 1.5$) were synthesized by changing the amount of solution B added to solution A. And composite catalysts are named as X-BaF₂-TiO₂, where X (0.35, 0.75 and 1.5) corresponds to the weight ratio of BaF₂ to TiO₂ in the material."

1. The weaknesses of the manuscript are related with some lack of experimental information and explanation, although discussion is ok, and with some presentation issues. Consequently, corrections are needed and authors should present the information in a more clear and precise way.

Response: Thank you for your help. According to your guidance, I have made corresponding corrections.

2. Page 3: - Line 21: "Composites" instead of "composites"

Response: Thank you very much. Done as suggested.

3. - Line 23-24: Authors may not generalize the catalytic activity for organic matter since they have just tested methyl orange solutions. The sentence needs to be rewritten, something as "has a good catalytic effect on degradation of methyl orange by gamma irradiation".

Response: Thank you very much. I have corrected all similar statements in the revised manuscript.

Eg 1: "It was demonstrated that the composite is found to be more efficient than prepared TiO₂ and commercial P25 in the degradation of methyl orange under γ -irradiation. "

Eg 2: "Overall, the BaF₂-TiO₂ composite material prepared herein is an excellent γ -irradiation degradation methyl orange catalyst."

4. Page 3: Line 30: a hyphen is missing.

Response: Thank you very much. Done as suggested.

5. Page 3:- Line 49: "Like traditional gamma-ray irradiation technology" at the beginning of the sentence does not make sense. Sentence may start at "However..."

Response: Thank you very much. In the re-written introduction, this sentence is corrected according to your instructions. As following:

"However, like conventional ultraviolet radiation and ionizing radiation, there are problems requiring large doses and long reaction times before producing sufficient active particles."

6. Page 4: Attention should be given to the 3.2 section.

- Line 30-31: "(5 mL concentrated hydrochloric acid)" is not needed and induce confusion

Response: Thank you very much. Done as suggested.

"Briefly, 60 mL of 1 mol L⁻¹ HCl was mixed with 7.5 g of citric acid in a crucible and stirred until the solution became clear."

7. Page 4: Line 42: Last sentence must include the explanation about the way authors refer to the pristine and obtained materials. For instance, it must be said here, and not just at page 9, that synthesized TiO₂ will be called "TiO₂" while "P25" will be used to designate the commercial TiO₂

obtained from Aladdin Chemical Reagent – this will facilitate the reading and comprehension of the manuscript.

Response: Thanks you for pointing this out. Done as suggested. In addition, inspired by you, I also renamed the composite catalyst.

"Catalysts with different BaF₂ and TiO₂ contents (i.e., C_{BaF₂}:C_{TiO₂} = 0.35, 0.75, 1.5) were synthesized by changing the amount of solution B added to solution A. And composite catalysts are named as X-BaF₂-TiO₂, where X (0.35, 0.75 and 1.5) corresponds to the weight ratio of BaF₂ to TiO₂ in the material. Additionally, neat TiO₂ and BaF₂ were obtained through a similar preparation method. In the following description, synthesized TiO₂ will be called "TiO₂" while "P25" will be used to designate the commercial TiO₂ obtained from Aladdin Chemical Reagent."

8. Page 4:- Please check Figure 1, some letters are missing. Also it legend should be changed, ex: Schematic representation of the preparation...

Response: Thanks you for pointing this out. Done as suggested.

Fig. 1 Schematic representation of the preparation the "BaF₂-TiO₂" composite"

9. Page 5: - Please indicate the version of software program ImagePro Plus.

Response: Thanks you for pointing this out. Done as suggested.

"The obtained SEM image was analyzed with the ImagePro Plus software 6.0(Media Cybernetics, Inc, Netherlands) program to determine the catalyst particle size."

10. Page 5:- Attention to section 3.4, please clarify: units of dose rate, why different dose rates and methyl orange concentrations were used at gamma irradiation and electron beam irradiations? This is important information that is not present, someone who are not familiar with the techniques might not understand it.

Response: Thanks you for pointing this out. I have been involved in the interpretation of this issue in Section 3.4, as following:

"In the experiment of electron beam irradiation degradation of methyl orange solution, since the electron beam energy emitted by the electron accelerator is 10 MeV in the experiment, the lowest absorbed dose after irradiation of the sample is 2 kGy, and the 20 mg/L methyl orange solution will completely degraded at the lowest dose, which does not reflect the ability of the catalyst to catalyze the degradation of methyl orange, so we increase the concentration of methyl orange solution to 50mg/L. And although we increase the solubility of the solution, it still belongs to the dilute aqueous solution, and does not affect the yield of unit dose of active particles, the experimental result is still very reliable."

11. Page 6: Authors state at 3.2 section that have studied different BAF₂ and TiO₂ contents but just present results for one ratio. Something about the other ration must be said (are they identical? The results shown represent all the ratios?).

Response: Thank you very much. Because the experiment is to test the effect of the composite catalyst on the degradation of methyl orange. In order to make the image more concise, in each experiment I only selected the composite catalyst with the best catalytic effect for explanation. However, in the revised manuscript, other proportions of the composite catalyst are also described in the text. As following

Eg 1: "XRD patterns of the prepared BaF₂, TiO₂, and BaF₂-TiO₂ samples are shown in Figures 2, respectively. BaF₂ is of Frankdicksonite type composition; its crystal form is cubic, as indicated by consistency with the standard diffraction peaks in JCPDS card No. 01-001-0533. The (101), (112), (200), (105), (211), (204), (116), (220), and (215) crystal faces of TiO₂ are all anatase crystal forms, as indicated by consistency with the standard diffraction peaks in JCPDS card No. 01-071-1167. The BaF₂-TiO₂ composites have almost all 2θ values of BaF₂ and TiO₂, but the peak intensities are different. It is observed that the peaks belonging to BaF₂ gradually increased with the increase of BaF₂ content for the composite catalysts. In addition, according to Equation (1), the crystallite sizes of BaF₂ and TiO₂ are 38 nm and 27 nm.

Fig. 2 XRD patterns of (a) BaF₂, (b) TiO₂, (c) BaF₂-TiO₂ (0.35-BaF₂-TiO₂) , (d) BaF₂-TiO₂ (0.75-BaF₂-TiO₂) and (e) BaF₂-TiO₂ (1.5-BaF₂-TiO₂) "

Eg 2:

"Ultraviolet photocatalytic degradation of methyl orange

Using a mercury lamp as the ultraviolet light source, the catalytic effects of P25, BaF₂, TiO₂, and 0.35-BaF₂-TiO₂ on the degradation of methyl orange solutions were investigated under the same conditions; the catalyst concentration was 1 g L⁻¹, the 0.35-BaF₂-TiO₂ concentration was 1.35 g

L^{-1} , and the TiO_2 concentration in the composite catalyst was 1 g L^{-1} . The UV visible spectrum and decolorization rate of the methyl orange solution after UV irradiation for 1 h are shown in Figs. 7 and 8. As seen, the decolorization rate of the methyl orange solution irradiated with pure ultraviolet light was only 4.93%, indicating that pure ultraviolet light has little effect on the degradation of methyl orange. With the addition of BaF_2 as a catalyst, the decolorization rate of methyl orange decreased to 3.25%, demonstrating that BaF_2 is not suitable for use in UV-catalyzed degradation of methyl orange. This may result because the absorption of ultraviolet light by the methyl orange solution is hindered when BaF_2 particles are distributed in the solution, resulting in a decrease in the decolorization rate of the methyl orange solution. By comparison, the decolorization rate of the methyl orange solution reached 47.64% and 48.78%, respectively, when TiO_2 and P25 were added, indicating an almost identical photocatalytic ability of these catalysts. Combined with the above characterization results, these results further demonstrate that our synthesized TiO_2 is our desired crystal form. When the 0.35- BaF_2 - TiO_2 composite was added as a catalyst, the decolorization rate of methyl orange solution was only 19.3%, which is lower than that of pure TiO_2 . A possible reason for this is limited penetration of the ultraviolet light into the solution. And the presence of BaF_2 hinders UV absorption by TiO_2 , resulting in some catalytic ability of the prepared composite sample but a rather weak photocatalytic ability. The theory is also supported by the catalytic ability of different composite catalysts in the experiment (The higher the BaF_2 content, the lower the catalytic effect).

Fig. 7 Ultraviolet-visible spectra before and after ultraviolet light irradiation of methyl orange solution with different catalysts (C_{BaF_2} , C_{TiO_2} , C_{P25} : 1 g L^{-1} ; $C_{0.35\text{-BaF}_2\text{-TiO}_2}$: 1.35 g L^{-1})

Fig. 8 Effect of different catalysts on the decolorization rate of methyl orange solutions irradiated with UV light (C_{BaF_2} , C_{TiO_2} , C_{P25} : 1 g L^{-1} ; $C_{0.35\text{-BaF}_2\text{-TiO}_2}$: 1.35 g L^{-1})"

Fig 3 "In addition, as the content of BaF₂ in the composite catalyst increases, the specific surface decreases, which may be one of the reasons for the decrease in the catalytic ability of 1.5-BaF₂-TiO₂. Therefore, we conclude that the optimum BaF₂-to-TiO₂ doping ratio is 0.75 ($C_{\text{BaF}_2}:C_{\text{TiO}_2}=0.75$)."

12. Page 7: - Lines 40-42: must be rewritten, it is not very clear

Response: Thank you very much. Done as suggested.

"Since the concentration of the catalyst in the next catalytic experiment is based on the concentration of TiO₂ (i.e., the concentration of TiO₂ in the composite catalyst added the same as the concentration of pure TiO₂), it is necessary to know how much composite catalyst is required per gram of TiO₂. Table 2 lists the total mass required when different composite catalysts contain 1g of TiO₂."

13. Page 8:- Table is not clear also, according to the legend the columns should exchange position

Response: Thank you very much. Because I renamed the composite catalyst, I made some changes to this table. As following:

"Table 2 Different composite catalysts Ti/Ba and mass of composite catalysts required for 1 g of TiO₂."

Catalyst	Ti/Ba	m _{BaF₂-TiO₂} (/g)
0.35-BaF ₂ -TiO ₂	6	1.35
0.75-BaF ₂ -TiO ₂	3	1.75
1.5-BaF ₂ -TiO ₂	1.5	2.5

14. Section 4.2 – 4.4:- The difference in the composite concentrations tested must be explained (not just stated).

Response: Thank you very much. Done as suggested. As following:

"Section 4.2: it is clear that the prepared composite catalyst has almost no catalytic ability for electron beam degradation of organic matter. The possible reason for this is that the energy of the electron beam is too high, beyond the absorption range of BaF₂ and TiO₂. Therefore, the addition of the catalyst in this experiment has almost no effect."

Section 4.3: This may be due to the fact that an appropriate amount of BaF₂ can increase the catalytic performance of the composite catalyst, while the excess BaF₂ blocking the TiO₂ surface.

That is, the excess BaF₂ affects the formations of •OH and defects on the TiO₂ surface that would catalytically degrade methyl orange molecules. In addition, as the content of BaF₂ in the composite catalyst increases, the specific surface decreases, which may be one of the reasons for the decrease in the catalytic ability of 1.5-BaF₂-TiO₂. Therefore, we conclude that the optimum BaF₂-to-TiO₂ doping ratio is 0.75 ($C_{\text{BaF}_2}:C_{\text{TiO}_2}=0.75$)

Section 4.4: Although the trend in the decolorization rate is similar to that when P25 is used as the catalyst, the BaF₂-TiO₂ composite demonstrates a superior catalytic effect. When the TiO₂ concentration in the composite catalyst is 1 g L⁻¹, the decolorization rate of the methyl orange solution is 52.24%, and when its concentration reaches the optimum value of 3 g L⁻¹, there are more catalysts in the solution, which will produce more active particles, and the decolorization rate increased to 79.38%. At this point, the catalyst concentration in the solution has reached saturation. Continued increase in concentration will hinder the absorption of gamma rays by the methyl orange solution, further affecting the yield of active particles produced by water radiolysis, and resulting in a decrease in catalytic effect.

4.5 Mechanism of composite catalyst

The mechanism of high energy radiation used to induce TiO₂-BaF₂ composite catalyst γ -radiation catalysis is not yet clear. It is true that composite catalyst can be excited by ⁶⁰Co irradiation source, thus interior UV from radioluminescence(RL) by radio-sensitive BaF₂ should be a possible route. When the composite catalyst was irradiated with γ -irradiation, Ba²⁺(5p) was excited to Ba²⁺(5p*) leaving a hole, and then electrons from F⁻(2p) valence band to the cation Ba²⁺(5p) level with the

release of 5.6 eV, 6.4 eV radiation(As shown in Equation 11 and Equation 12)[17]. In addition, an interior electric field developed in BaF₂-TiO₂ depletion layer is another probably scheme. It is believed that the formed TiO₂ particles are probably combined with the BaF₂ surface via the Ti–O–Ba structural units .Since the energy band structures of TiO₂ and BaF₂ are different from each other, a typical “hetero-junction”(HJ) would be formed between TiO₂ and BaF₂. Due to the energy bands of TiO₂ and BaF₂ will bend into each other within this hetero-junction that benefit charge separation within composite catalysts. An inner electronic fields thus established in the hetero-junction directed from BaF₂ to TiO₂. Under this inner electric field, radiationinduced electrons in the TiO₂ will drift into the BaF₂ to endow the composite with irradiation -catalytic activity(As shown in Equation 13) [20-22]. We believe that the γ -radiation catalytic mechanism of composite catalyst happens by near BaF₂ and TiO₂ and seemed to hybridizing of γ -irradiation and UV as illustrated in Figure 15.

Fig. 15 Scheme of the mechanism of BaF₂-TiO₂ γ -irradiation catalytic reaction"

15. - Ratio $m\text{BaF}_2 : m\text{TiO}_2$ should be clarified and also presented at the legends of figures 5, 7, 9 and 13.

Response: Thank you very much. Done as suggested. In addition, due to the addition of some content, the number of the picture has also changed. As following:

"Fig. 7 Ultraviolet-visible spectra before and after ultraviolet light irradiation of methyl orange solution with different catalysts (C_{BaF_2} , C_{TiO_2} , C_{P25} : 1 g L^{-1} ; $C_{0.35\text{-BaF}_2\text{-TiO}_2}$: 1.35 g L^{-1}) "

"Fig. 9 Ultraviolet-visible spectra before and after γ -ray irradiation of methyl orange solutions with different catalysts (C_{BaF_2} , C_{TiO_2} , C_{P25} : 1 g L^{-1} ; $C_{0.75\text{-BaF}_2\text{-TiO}_2}$: 1.75 g L^{-1}) "

"Fig. 11 Ultraviolet-visible spectra before and after electron beam irradiation of methyl orange solutions with different catalysts (C_{BaF_2} , C_{P25} : 1 g L^{-1} ; $C_{0.75\text{-BaF}_2\text{-TiO}_2}$: 1.75 g L^{-1}) "

"Fig. 12 Effect of different catalysts on the decolorization rate of methyl orange solutions irradiated with an EB (C_{BaF_2} , C_{P25} : 1 g L^{-1} ; $C_{0.75\text{-BaF}_2\text{-TiO}_2}$: 1.75 g L^{-1}) "

16. - The value of decolorization rate obtained with $\text{BaF}_2\text{-TiO}_2$ at Figure 8 does not match any of those presented at Figure 11! Why?

Response: Thanks you for pointing this out. I made a mistake while processing the data. I have corrected this error in the revised manuscript. In addition, since I added some pictures, the number of the original picture has also changed. As following:

Fig. 10 Effect of different catalysts on the decolorization rate of γ -ray irradiated methyl orange solutions

(C_{BaF_2} , C_{TiO_2} , C_{P25} : 1 g L⁻¹; $C_{\text{0.75-BaF}_2\text{-TiO}_2}$: 1.75 g L⁻¹)

17. - At section 4.4 authors mentioned the effect of TiO₂ as catalyst but graphical information at Figure 12 b shows P25. Please correct.

Response: Thank you very much. Done as suggested.

18. Authors may not generalize the catalytic activity for organic matter since they have just tested methyl orange solutions. Please correct.

Response: Thank you very much. Done as suggested.

"Therefore, more active particles capable of degrading methyl orange are produced. "

"Overall, the BaF₂-TiO₂ composite material prepared herein is an excellent γ -irradiation degradation methyl orange catalyst. "

Special thanks to you for your good comments.

I tried my best to improve the manuscript and made some changes in the manuscript. These changes will not influence the content and framework of the paper.

I appreciate for editors and reviewers' warm work earnestly, and hope that the correction will meet with approval.

Once again, thank you very much for your comments and suggestions.

Thank you and best regards.